# Improving Wind and Power Predictions via Four-Dimensional Data Assimilation in the WRF Model: Case Study of Storms in February 2022 at Belgian Offshore Wind Farms

Tsvetelina Ivanova[1,2], Sara Porchetta[3,4], Sophia Buckingham[5], Gertjan Glabeke[1,6], Jeroen van Beeck[1], and Wim Munters[1]

[1]Environmental and Applied Fluid Dynamics Department, von Karman Institute for Fluid Dynamics, Chau. de Waterloo 72, 1640 Rhode-Saint-Genèse, Belgium

[2]Department of Engineering Technology, Vrije Universiteit Brussel, Pleinlaan 2, 1050 Brussels, Belgium

[3]Civil and Environmental Engineering, Massachusetts Institute of Technology, 77 Massachusetts Avenue, 1-290, Cambridge, MA 02139, United States of America

[4]Department of Geoscience and Remote Sensing, Faculty of Civil Engineering and Geosciences, Delft University of Technology, Stevinweg 1, 2628 CN Delft, Netherlands

[5]Research & Innovation, ENGIE Laborelec, Rodestraat 125, 1630 Linkebeek, Belgium

[6]Department of Civil Engineering, Hydraulics Laboratory, Ghent University, Sint-Pietersnieuwstraat 41, B-9000 Gent, Belgium

**Correspondence:** Tsvetelina Ivanova (tsvetelina.ivanova@vki.ac.be)

**Abstract.** Accurate modeling of wind conditions is vital for the efficient operation and management of wind farms. This study investigates the enhancement of weather simulations by assimilating local offshore LiDAR and/or SCADA data into a numerical weather prediction model, while considering the presence of neighboring wind farms through wind farm parameterization. We focus on improving model output during storms impacting the Belgian-Dutch wind farm cluster located in the Southern Bight of the North Sea via the four-dimensional data assimilation (nudging) technique in the WRF model. Our findings indicate that assimilating wind observations significantly reduces the relative root-mean-square error for wind speed at a wind farm located $47\,\mathrm{km}$ downwind from the offshore LiDAR platform. This leads to more accurate power production outputs. Specifically, at wind turbines experiencing wake effects, the wind speed error decreased from $10.5\,\%$ to $5.2\,\%$, and the wind direction error was reduced by a factor of $2.4$. A proposed artificial configuration, leveraging the upwind LiDAR measurements, showcases the potential for improving hour-ahead wind and power predictions. Moreover, we perform a thorough sensitivity study to nudging parameters during versatile atmospheric conditions, which helps to identify best assimilation practices for this offshore setting. These insights are expected to refine wind resource mapping and reanalysis of weather events, as well as to motivate more measurement campaigns offshore.

## 1 Introduction

In recent years, wind energy has emerged as a crucial and rapidly growing renewable energy source. Accurate numerical simulation of wind speed, wind direction, and power production has become essential for efficient planning, design, and

operation of wind farms. The use of numerical weather prediction (NWP) models, such as the open-source Weather Research and Forecasting Model (WRF) (Skamarock et al., 2019), developed at the National Center for Atmospheric Research (NCAR), plays a vital role in obtaining these accurate model outputs.

For weather simulations at offshore wind farm zones, NWP models do not necessarily have sufficiently accurate initial conditions due to the sparsity of offshore observations. This lack of data leads to broader issues, where NWP models can display large bias errors due to the overall lack of long-term offshore measurement data (Archer et al., 2014). Furthermore, extreme events in which wind turbines are still operating can have a profound impact on wind farm operations. Such events involving high wind speeds can often lead to implications for power generation and grid stability. The impact at near-ground

levels can be damaging for human activity, as well as cause grid instabilities and potential wind turbine cut-outs. This further increases the need for accurate weather simulations. Numerous works are dedicated to studying and understanding extreme events, such as Larsén et al. (2019), Pryor and Barthelmie (2021), Vemuri et al. (2022), Sethunadh et al. (2023). Given that extreme events are often influenced by the larger-scale dynamics of the atmosphere, NWP models are commonly employed to analyze and predict them. However, accurately capturing extreme events that significantly impact wind energy remains a

challenge.

Improving wind and power simulations extends beyond model settings, and can be enhanced by incorporating additional physics. One such path to consider is the impact of wind farms on the atmosphere. In NWP models, wind farms can be represented by a wind farm parameterization (WFP). Over the years, different WFPs have been proposed. By employing WFP, the influence of wind farms on the surrounding atmospheric conditions is accounted for, and consequently, an insight into

approximated inter-farm dynamics is possible. A study by Lee and Lundquist (2017) quantifies wind and power prediction improvements that are achieved by incorporating WFP. A systematic literature review by Fischereit et al. (2022a) compares 10 existing WFPs. Furthermore, Fischereit et al. (2022b) highlights that the WFP of Fitch et al. (2012) is a suitable state-of-the-art choice for modeling the presence of wind farms in WRF, and is selected in the present work. This WFP models the wind farm as a momentum sink and a turbulent kinetic energy (TKE) source. Other WFP applications include wind-wave coupling

studies, such as Porchetta et al. (2021). Accurately capturing atmospheric conditions is crucial for modeling wind farm wakes. However, when representing wind farm wakes in mesoscale models using WFP, uncertainties arise (Eriksson et al., 2017; Peña et al., 2022; Ali et al., 2023). WFP has limitations, including the need for TKE correction, as well as its sensitivity to atmospheric stability. Additionally, WFPs restrict options for planetary boundary layer schemes in NWP models, which in turn affects the fidelity of boundary layer representation.

Along with model setting and including more physics within WRF, improvements in wind and power model output can also be achieved by employing data assimilation (DA) techniques. Data assimilation is the process of integrating observed data into a numerical model (Skamarock et al., 2019). We distinguish between two groups of such techniques: variational DA (Barker et al., 2012) and four-dimensional data assimilation (FDDA or nudging, Liu et al. (2008)). Variational DA is concerned with finding the optimal initial state of the atmosphere (in the case of three-dimensional variational data assimilation (3DVar), Barker

et al. (2004)), and furthermore, with finding the optimal model trajectory based on this optimal initial state (in the case of four-dimensional variational data assimilation (4DVar), Huang et al. (2009); Zhang et al. (2013, 2014)). Both 3D and 4D variational

DA techniques rely on minimizing the difference between model forecasts and observations by optimizing a cost function. One work that exploits the benefits of variational data assimilation is by Sun et al. (2022), in which wind speed forecasts are improved when assimilating observations from the nacelle of turbines. In contrast, FDDA (nudging) operates differently from variational data assimilation: FDDA directly influences the state variables over time in order to match observed data (Reen, 2016), and it is the selected method in this work. In FDDA, the approach is to introduce tendency terms in the model equations to adjust the prognostic variables, such as temperature, humidity, and wind components, towards observed values. This approach acts as a controller, rather than a cost function optimizer. This makes FDDA much more computationally efficient than variational methods, and this is highly relevant in an operational context (Cheng et al., 2017). A drawback of this method is that only prognostic model variables can be assimilated. Besides observational FDDA, it is also possible to perform grid nudging and spectral nudging in WRF: these are out of the scope of this work. Several studies have explored the leverage of data assimilation techniques in mesoscale models for wind energy applications. For example, Kosovic et al. (2020) use RTFDDA (real-time FDDA) for local data, integrated with a machine learning approach for power estimation. Nudging techniques are also applied in the onshore study of Cheng et al. (2017) that highlights the effectiveness of RTFDDA (within a customized version of WRF) in improving wind energy predictions 0–3 hours ahead for normal weather conditions, using only wind speed observations from wind turbine anemometers. Furthermore, the study of Mylonas et al. (2018) performs FDDA of observations from the offshore meteorological mast FINO3 in the North Sea for wind resource assessment and reanalysis.

Our research presents a novel approach to improving wind and power model output by integrating the advantages of a physics-based WFP and FDDA in WRF, particularly during extreme offshore conditions. We focus on utilizing observational FDDA of horizontal wind components gathered from a LiDAR (Light Detection and Ranging) vertical profiler. This approach is unique in its offshore application of FDDA in WRF due to the strategic placement of the LiDAR upstream (with respect to the most common South-Westerly winds in the Southern Bight of the North Sea). This geographical advantage allows for advanced information on incoming wind conditions to be provided approximately one hour ahead, which is the advective time required. Our goal is to improve the accuracy of wind and power model output offshore by assimilating this data during significant events such as the storms Eunice and Franklin in February 2022 over the Belgian North Sea. These events had a substantial impact on wind power production (reported, for example, in Belgian Offshore Platform News (2022)), making them crucial case studies for our research. Furthermore, to gain insight into optimal FDDA settings for this offshore configuration, we experiment with sensitivity to different observational nudging parameters, such as nudging strength and horizontal radius of influence of the assimilated observations. We evaluate the performance of the simulations by comparing the results to LiDAR and SCADA (Supervisory Control And Data Acquisition) datasets, using classic metrics from the state-of-the-art handbook on wind forecasting by Yang et al. (2021). These metrics are MAE (mean absolute error), RMSE (root-mean-square error), and bias with respect to observations.

The paper is structured as follows. Section 2 describes the methodology and the configuration of the numerical setup of the WRF model including the FDDA algorithm, the available offshore observations, and selected case studies for simulations in this work. Section 3 expresses the results and discussion for different simulation scenarios. Finally, Sect. 4 outlines the conclusions of this paper.

## 2 Methodology and numerical setup

The NWP model employed in this work is the Advanced Research WRF (ARW) Model (Skamarock and Klemp, 2008; Skamarock et al., 2019), version $4.5.1$, which is a state-of-the-art mesoscale NWP system available in the public domain. It solves the fully compressible non-hydrostatic Euler equations, and it has a rich set of physics parameterizations.

### 2.1 The WRF model configuration

Our study is focused on the Belgian-Dutch wind farm cluster. The setup consists of five nested domains, three of which are identical and innermost. The domains have their names and grid cells as follows. D01: $150 \times 150$; D02: $190 \times 190$; D03, D04, and D05: $220 \times 190$ grid cells; centered at latitude $51.42°$ N and longitude $2.74°$ E, with one-way nesting. The Lambert conformal projection is selected. The three identical innermost domains D03, D04, and D05, are of interest, with a size of $680$ km by $596$ km. The horizontal grid spacing is $18$ km for the outer domain, $6$ km for the intermediate domain, and $2$ km for the three innermost domains. The latter follows the guidelines of Fischereit et al. (2022a) to use horizontal grid spacing of at least 3 to 5 times the wind turbine rotor diameter for the domains with active WFP (in this case, D04 and D05). The model configurations in the three innermost domains (with $2$ km grid spacing) differ in the following way: D03 is for simulations without WFP, D04 is for active WFP, and D05 – for active WFP while performing FDDA. The domains are shown in Fig. 1, along with key measurement locations of three LiDARs: at the Westhinder (WHi) platform (Glabeke et al., 2023), as well as at the Lichteiland Goeree platform (LEG) and Europlatform (EPL) (Wind@Sea project, Wind Energy Research Group at TNO Energy Transition, 2023).

The setup in this work is based on Hahmann et al. (2020), Dörenkämper et al. (2020), and Larsén and Fischereit (2021). Relevant details on the parameterizations used in the setup can be found in Table 1. The cumulus scheme is used only on the outermost domain. Our study involves usage of the WFP of Fitch et al. (2012), which further requires the introduction of sufficient vertical model levels. This allows a representative description of the wind profile across the rotor, which is done by relying on recommendations from Lee and Lundquist (2017). The vertical levels are stretched, which ensures more levels close to the surface. The total number of levels is $80$ (Lee and Lundquist, 2017). The lowest level is at $6$ m, with sufficient points across the wind turbine rotor for a typical offshore wind turbine (WT) in the Belgian-Dutch cluster. More specifically, for the smallest wind turbine within the Belgian side of the Belgian-Dutch cluster, there are 8 vertical levels that span across the rotor, whereas for the largest wind turbine of this cluster, there are 15 levels. The model pressure top is at 1000 Pa.

To consider the impact of wind farms, we incorporate not only the Belgian-Dutch wind farm cluster, but also the fully-commissioned offshore wind farms in the South Bight of the North Sea via the WFP of Fitch et al. (2012) within the WRF model. This group of wind farms consists of $1409$ wind turbines (out of $5779$ in total in Europe and UK (Hoeser et al., 2022)) within 27 different wind farms that are represented in the setup, in proximity to the Belgian-Dutch cluster. The locations of these wind farms are extracted from Hoeser et al. (2022), and the publicly available dataset of Hoeser and Kuenzer (2022). The details of the different wind farms are summarized in Appendix A, Table A1. Besides wind turbine locations, the WFP requires the power and thrust curves for each wind turbine in order to simulate its effects on the atmosphere. These were obtained

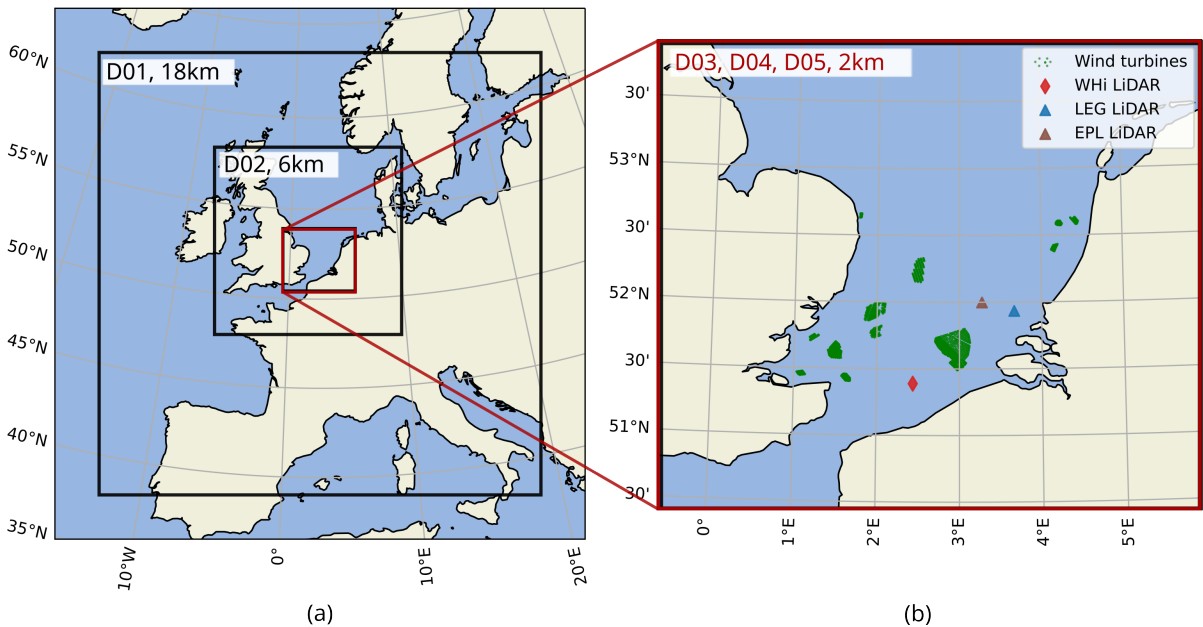

**Figure 1.** Nested domains in WRF (a). The domains of interest (b) are the three identical D03, D04, D05, with a grid spacing of $2\,\mathrm{km}$. Key measurement locations are indicated (i.e. WHi LiDAR, LEG and EPL LiDARs). The 1409 wind turbines that are currently included in the WRF setup are also visualized in (b).

from a mix of both open (https://www.thewindpower.net/; and WindPRO (EMD-International)), and confidential sources. The performed WFP simulations take into account the TKE advection with a correction factor $\alpha$ of 0.25, following Archer et al. (2020). This coefficient $\alpha$ is denoted in Table 1.

For time integration, a third order Runge-Kutta scheme is used, and for advection – second- to sixth-order spatial discretization schemes. For the model integration, adaptive time stepping is used, with a target Courant–Friedrichs–Lewy number of 0.6. For initial and boundary conditions, the Global Forecast System (GFS) 3-hourly data from NCEP's Historical Archive is used, with forecast grids on a 0.25 by 0.25 global latitude longitude grid (National Centers for Environmental Prediction, National Weather Service, NOAA, U.S. Department of Commerce, 2015). All simulated periods include a 12-hour spin-up time. These periods of interest are detailed in Sect. 2.3 and include extreme events in February 2022 (storms Eunice and Franklin).

Our study investigates the sensitivity of results to nudging parameters. Varying such parameters helps to gain insight into the most suitable assimilation strategies for this offshore configuration. For a selected day (17 February 2022), a number of simulations are performed by varying the horizontal radius of influence $R_{xy}$, as well as the nudging strength $G_q$. In these numerical experiments, either LiDAR or SCADA data is assimilated. The FDDA algorithm is detailed in Sect. 2.2. The full sensitivity experiment is described in Sect. 2.5.

**Table 1.** Parameterization options and configuration in the present WRF setup

| Parameterization | Scheme (with option in namelist) | Reference |
|---|---|---|
| PBL scheme | MYNN 2.5 level TKE (option 5) | Nakanishi and Niino (2006) |
| Cumulus | Kain-Fritsch (option 1) | Kain (2004) |
| Microphysics scheme | Thompson et al. (option 8) | Thompson et al. (2008) |
| Radiation | RRTMG (option 4) | Iacono et al. (2008) |
| Land surface model | NOAH LSM (option 2) | Ek et al. (2003) |
| Wind farm parameterizaiton | when active: Fitch (option 1) | Fitch et al. (2012) |
| TKE correction coefficient $\alpha$ | 0.25 (default value) | Archer et al. (2020) |

## 2.2 The FDDA (nudging) algorithm

WRF has an implemented algorithm to assimilate prognostic model variables such as the horizontal components of wind speed via the FDDA technique, as described in Skamarock et al. (2019) and Reen (2016). With this algorithm, the numerical solution is nudged towards observations by introducing tendency terms in the model equations as

$$\frac{\partial q\mu}{\partial t}(x,y,z,t) = F_q(x,y,z,t) + \mu G_q \frac{\sum_{i=1}^{N} W_q^2(i,x,y,z,t)(q_o(i) - q_m(x_i,y_i,z_i,t))}{\sum_{i=1}^{N} W_q(i,x,y,z,t)},$$

where $q$ is the quantity being nudged (in this work, horizontal wind components that are projected from wind speed and wind

direction observations, as in Cheng et al. (2017)), $\mu$ is the dry hydrostatic pressure, $F_q$ are the physical tendency (or model forcing) terms of $q$, $G_q$ is the nudging strength, $N$ is the total number of observations, $W_q$ is the weighing function in space and time, $q_o$ is the observed value of the quantity of interest, and $q(x_i,y_i,z_i,t)$ is its model value. The working principle is of a proportional controller: with the approaching of the model value to its observed value, the nudging tendency term decreases.

The weighing function $W_q$ can be expressed as the product of horizontal ($w_{xy}$), vertical ($w_\sigma$), and temporal ($w_t$) contribu-

145 tions (Xu et al., 2002). The contribution $w_{xy}$ is a function of the horizontal radius of influence $R_{xy}$, $w_\sigma$ is the vertical weighing function, and $w_t$ is a function of the assimilation time window $\tau$ over which an observation is used in the nudging algorithm. The horizontal weighing function is a Cressman-type function given by

$$w_{xy} = \frac{R_{xy}^2 - D^2}{R_{xy}^2 + D^2}, \quad 0 \le D \le R_{xy},$$
$$w_{xy} = 0, \qquad D > R_{xy},$$

where $R_{xy}$ is the radius of influence and $D$ is the distance from the observation location to the grid point. The vertical and the temporal weighing functions are also distance-weighted. Further details on observational nudging can be found in Grell et al. (1994) and in Xu et al. (2002).

In this work, we perform a sensitivity study to the horizontal radius of influence $R_{xy}$, as well as to the nudging strength $G_q$. This sensitivity study is described in Sect. 2.5 with all tested nudging parameter values.

## 2.3 Available offshore observations

In this section, we detail all offshore observations that are used in this work, both for assimilation in WRF, and for simulation performance evaluation. The locations of all offshore observations are indicated in Fig. 2(a). These datasets are collected at five different locations: by three LiDAR profilers (at the Westhinder platform (WHi), at the Lichteiland Goeree (LEG) platform, and at the Europlatform (EPL)), as well as by two groups of wind turbines (SCADA from nacelle anemometers at Front & Waked WTs).

**LiDAR profiler at the Westhinder platform (WHi)**

The observations used for assimilation are collected by a vertical LiDAR profiler at the Westhinder survey platform (with coordinates 51°23'18.74"N, 02°26'16.18"E). The vertical profiling LiDAR (ZX 300M) is shown in Fig. 2(b). It has been installed at the Westhinder platform since August 2021 and has been collecting wind speed and wind direction information since (Glabeke et al., 2023). In this study, we utilize WHi LiDAR data that is available from 4 August 2021 to 18 July 2022, as well as from 26 January to 6 February 2023. This LiDAR measures at 11 different heights (34.5, 44.5, 62.5, 79.5, 104.5, 124.5, 149.5, 174.5, 224.5, 274.5 and 324.5 mTAW), as shown in Fig. 2(c). The measurement heights are in mTAW (meters Tweede Algemene Waterpassing), which means that the average sea level at low tide in Ostend (Belgium) is used as the zero level. This value for Ostend is $\pm 2.3$ m (positive and negative deviations) with respect to the mean sea level. The LiDAR is retrieving wind vector data at a frequency of 1 Hz per height. A full measurement of the vertical profile typically takes 17 seconds due to the extra time for beam focus adjustment, as well as for additional weather condition measurements used in quality control. Thus, for a 10-minute interval, the maximum number of validated wind speed and wind direction measurements is 35. The validation is based on a wind-industry filtering, performed by the LiDAR software (User's Manual ZephIR, 2018), as meteorological conditions can result in non-validated wind data (for example, due to low cloud ceiling, fog, or precipitation). The filtering criteria are selected based on a DNV (Det Norske Verita) classification.

The location in which the WHi LiDAR observations are collected is especially favorable, since the measurements are of free-stream wind, given the predominant South-Westerly winds (shown in the year-long wind rose in Fig. 2(d)). This allows information to propagate towards the farm of interest when performing FDDA of these local observations. The typical advection timescale of this propagation is approximately one hour. Therefore, the assimilation of such upwind data can help improve hour-ahead predictions. This approach is discussed in Sect. 2.5, and the results – in Sect. 3.1.

**SCADA from nacelle anemometers at Front & Waked WTs**

The nacelle anemometers gather in-situ data on horizontal wind speed and wind direction at the wind farms. Additionally, the SCADA system records power production data. For the purpose of our simulations, we have specifically selected two locations within the Belgian-Dutch cluster (which comprises 572 wind turbines) as depicted in Fig. 2. The first location, referred to as "Front WTs", includes a subset of five wind turbines. These turbines are strategically positioned in the front row, aligning with the most common wind direction from the South-West. This alignment is consistent both for the period under investigation

and for the overall prevailing wind direction in the Belgian North Sea, as shown from the wind rose in Fig. 2(d). The second location, "Waked WTs", consists of another subset of five wind turbines. These turbines are situated in the wake, on an arbitrary back row, of a selected Belgian wind farm (when the wind direction is from the South-West). The selection of these two distinct locations allows us to observe the effect of the wind farm parameterization across a few kilometers. It also enables the assessment of the area of impact of the data assimilation upwind.

The reason for selecting exactly five turbines per location (Front WTs and Waked WTs) is the computational domain of our simulations. Each subset of turbines is contained within a specific computational cell with a grid spacing of 2 km. Therefore, for both locations, we consider the average values from the SCADA of the corresponding five turbines, providing us with a representative sample for each computational cell. However, due to a non-disclosure agreement with the wind farm operator, we are unable to list the exact coordinates of these turbines.

**LiDAR at the Lichteiland Goeree (LEG) platform**

To further evaluate our numerical results, we compare them additionally to a LiDAR profiler on the Lichteiland Goeree (LEG) platform (coordinates $51°55'30"$ N, $3°40'12"$ E) provided by the Wind@Sea project, Wind Energy Research Group at TNO Energy Transition (2023) (https://www.tno.nl/, https://nimbus.windopzee.net/). The LEG platform collects meteorological observations and is positioned approximately 110 km away from the Westhinder platform, and approximately 63 km away from the Waked WTs location, as shown in Fig. 2(a). Wind speed observations are obtained via a Leosphere Windcube LiDAR V2.1 which can measure up to approximately 240 m above sea level (8 different heights at 62, 90, 115, 140, 165, 190, 215 and 240 m above sea level) simultaneously with a wind vector data rate of 1 Hz.

**LiDAR at the Europlatform (EPL)**

We perform comparisons of simulations with one more LiDAR ZX-300M wind profiler dataset that is collected at the Europlatform (EPL) also by the Wind@Sea project, Wind Energy Research Group at TNO Energy Transition (2023). The measurement heights of the EPL LiDAR are 63, 91, 116, 141, 166, 191, 216, 241, 266 and 291 meters. This platform is located in proximity to the LEG platform, as indicated in Fig. 2(a). The approximate distance between EPL and Waked WTs is 56 km.

**Performance metrics**

The WHi LiDAR and SCADA datasets (at Front & Waked WTs locations) are utilized for assimilation (nudging) in WRF in distinctive numerical experiments (described in Sect. 2.5), as well as for model output evaluation. Before assimilation in WRF, all wind speed and wind direction observations are projected onto the axes aligned with model $U$ and $V$ velocity variables (Cheng et al., 2017). The results obtained from simulations are compared to the five locations in total, shown in Fig. 2: three LiDARs (WHi, LEG, and EPL), as well as two locations (Front & Waked WTs) with wind turbine data (local wind speed, wind direction, and power) from a SCADA database.

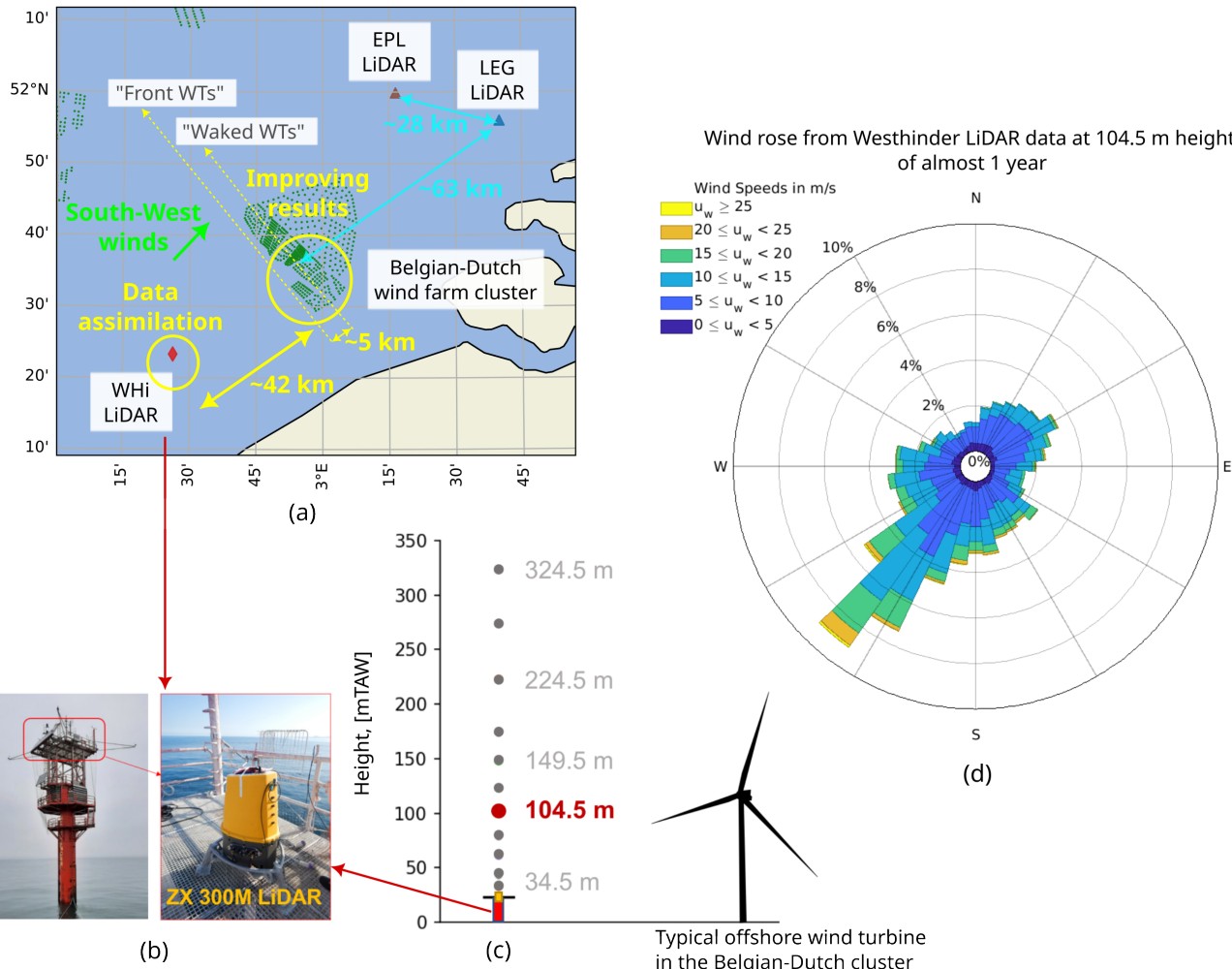

**Figure 2.** Locations of interest (a) within the innermost domains: the (typically) upstream WHi LiDAR; the Front WTs and Waked WTs from the Belgian-Dutch cluster; and the LEG and EPL LiDARs. The WHi LiDAR is shown on its platform in (b), and its illustration with respect to a typical wind turbine is shown in (c). Finally, a wind rose (d) obtained from the LiDAR dataset of Glabeke et al. (2023) at 104.5 mTAW for a period of almost one year (4 August 2021 to 18 July 2022).

To evaluate the performance of simulations, we utilize established metrics, outlined in Yang et al. (2021). These metrics include Mean Absolute Error (MAE) (Lydia et al., 2014), Root-Mean-Square Error (RMSE) (Zhao et al., 2011), and Bias (Wang et al., 2019). Each of these metrics provides a different perspective on the accuracy of our simulations compared to offshore observations. Let us denote the $i$th (normalized) model output variable as $p_m^i$, the $i$th (normalized) observed variable as $p_o^i$, and the length of the data set as N. The MAE is a widely used metric for evaluating wind model output, as it reflects the

**Table 2.** Time frames of interest in this study along with their corresponding goals.

| Label | (Duration) Time frame dates | Goal under specific conditions |
|---|---|---|
| F1 | (4 days) 17 February 2022 – 21 February 2022 | Improve model performance via FDDA of local upwind LiDAR data at near-hub height |
| F2 | (1 day) 17 February 2022 – 18 February 2022 | Sensitivity study of nudging strength and horizontal radius of influence to identify optimal FDDA practices |
| F3 | (2 days) 20 October 2021 – 22 October 2021 | Improve model performance via FDDA for a negative wind speed bias (when WFP is inactive) |
| F4 | (4 days) 30 January 2023 – 3 February 2023 | Improve model performance via FDDA for a positive wind speed bias (when WFP is inactive), including notable wind speed fluctuations. |

overall error level. It is calculated as the average absolute difference between the model output and the observed data:

$$\text{MAE} = \frac{1}{N} \sum_{i=1}^{N} |p_m^i - p_o^i|.$$

The RMSE is another important metric, particularly due to its sensitivity to outliers in NWP model output. It is computed as
the square root of the average squared differences between the model output and the observed data:

$$\text{RMSE} = \sqrt{\frac{1}{N} \sum_{i=1}^{N} (p_m^i - p_o^i)^2}.$$

Lastly, the bias indicates the average deviation of the model output from the actual observed values:

$$\text{Bias} = \frac{1}{N} \sum_{i=1}^{N} (p_m^i - p_o^i).$$

These metrics allow a comprehensive analysis and evaluation of model performance across different simulation scenarios.

**2.4   Case studies of different weather conditions**

The case studies of interest are described in Table 2. For an easier reference throughout the text, these cases are labeled as time frames F1 to F4. All four cases have a suitable South-West wind direction that allows the advection of the assimilated LiDAR data towards the Belgian-Dutch cluster (Fig. 2(a)). Each case study has a targeted goal. F1 enhances model accuracy using FDDA. F2 involves a comprehensive sensitivity analysis to nudging parameters, which is one of the main goals of this work.
F3 displays a negative wind speed bias when WFP is inactive, a condition that deteriorates with the activation of WFP due to further momentum extraction from an already underpredicting baseline. Conversely, F4 displays a positive wind speed bias when WFP is inactive, which is reduced once WFP is activated. Thus, for F1, F3, and F4, incorporating FDDA is explored to enhance model output. All times indicated throughout the text and figures are in UTC time zone.

Time frame F1 includes two storms (Eunice and Franklin) in February 2022. The goal within F1 is to use FDDA (of upwind LiDAR) to help improve model performance. For F1 at the WHi LiDAR location, according to the observations in Fig. 3 at 104.5 mTAW height, the storms have the following characteristics:

– Storm Eunice occurred from the early morning of 18 February to the early morning of 19 February. It reached a peak velocity of $37.13 \text{ m s}^{-1}$ at 14:00 on 18 February 2022, with the wind direction varying between 225 and 275 degrees.

– Storm Franklin spanned from the afternoon of 20 February to noon on 21 February. The peak velocity recorded up to $30.46 \text{ m s}^{-1}$ at 19:30 on 20 February 2022. Notably, both the wind speed and direction underwent significant changes, especially from 20:00 to 20:30 on 20 February 2022.

The selection of the F1 period is strategic, given its versatile atmospheric conditions. This period includes instances with wind speeds around the cut-out value, as well as times featuring rapid changes in wind direction. Four-dimensional data assimilation (nudging) is favorable whenever the wind direction is predominantly from South-West, as this allows for the Westhinder LiDAR to be upwind from the Belgian-Dutch cluster. This is indeed most often the case in the Southern Bight of the North Sea, as shown in the wind rose in Fig. 2(d). The average wind direction for F1 is West-Southwest. Note that gaps in the time series in Fig. 3 are due to data filtering performed by the LiDAR software based on meteorological conditions.

Time frame F2 is a selected day from F1 (17 February 2022), in which we focus on a sensitivity study of the nudging strength and radius of influence to identify optimal FDDA practices. This procedure is detailed in Sect. 2.5. The goal in F2 is to understand how different parameters of the FDDA algorithm impact model performance and accuracy, and to identify best FDDA practices for the offshore setting.

Time frame F3 spans over two days, from 20 October 2021 to 22 October 2021. During this period, we aim to demonstrate the capabilities of a selected FDDA setting (based on the analysis outlined in Sect. 2.5) for a case with a negative bias, specifically when WFP is inactive. This negative bias is even more pronounced when WFP is active (due to the momentum extracted from the flow), making F3 a useful case to test the performance of a selected FDDA setting. Finally, time frame F4 spans across four days, from 30 January 2023 to 3 February 2023. The goal during this period is to demonstrate the capabilities of the selected FDDA setting for a case with a positive bias, again when WFP is inactive. Additionally, this period also involves significant changes in wind speed data. This is identified using a straightforward SCADA data filtering approach, in which a time period is of interest if the difference of two consecutive wind speed data points (5 minutes apart) is larger than the corresponding standard deviation of $3.93 \text{ m s}^{-1}$. This allows testing the robustness of the selected FDDA setting under rapidly changing wind conditions.

In each of the time frames (F1, F2, F3, F4), we consistently perform two baseline simulations. The first simulation, referred to as 'WFP_off' in subsequent figures, does not account for the presence of wind farms within the computational domain. The second simulation, labeled 'WFP', incorporates the effects of these wind farms via the Fitch WFP. Within time frame F2, we perform an extensive set of 20 numerical experiments to assess the sensitivity of results to varying nudging parameters. The details of this study are elaborated in Sect. 2.5. This helps to identify a preferred (optimal) nudging setting. This setting is then applied to the other case studies F1, F3, F4 (labeled 'WFP FDDA'). For time frame F1, we conduct a comparative analysis

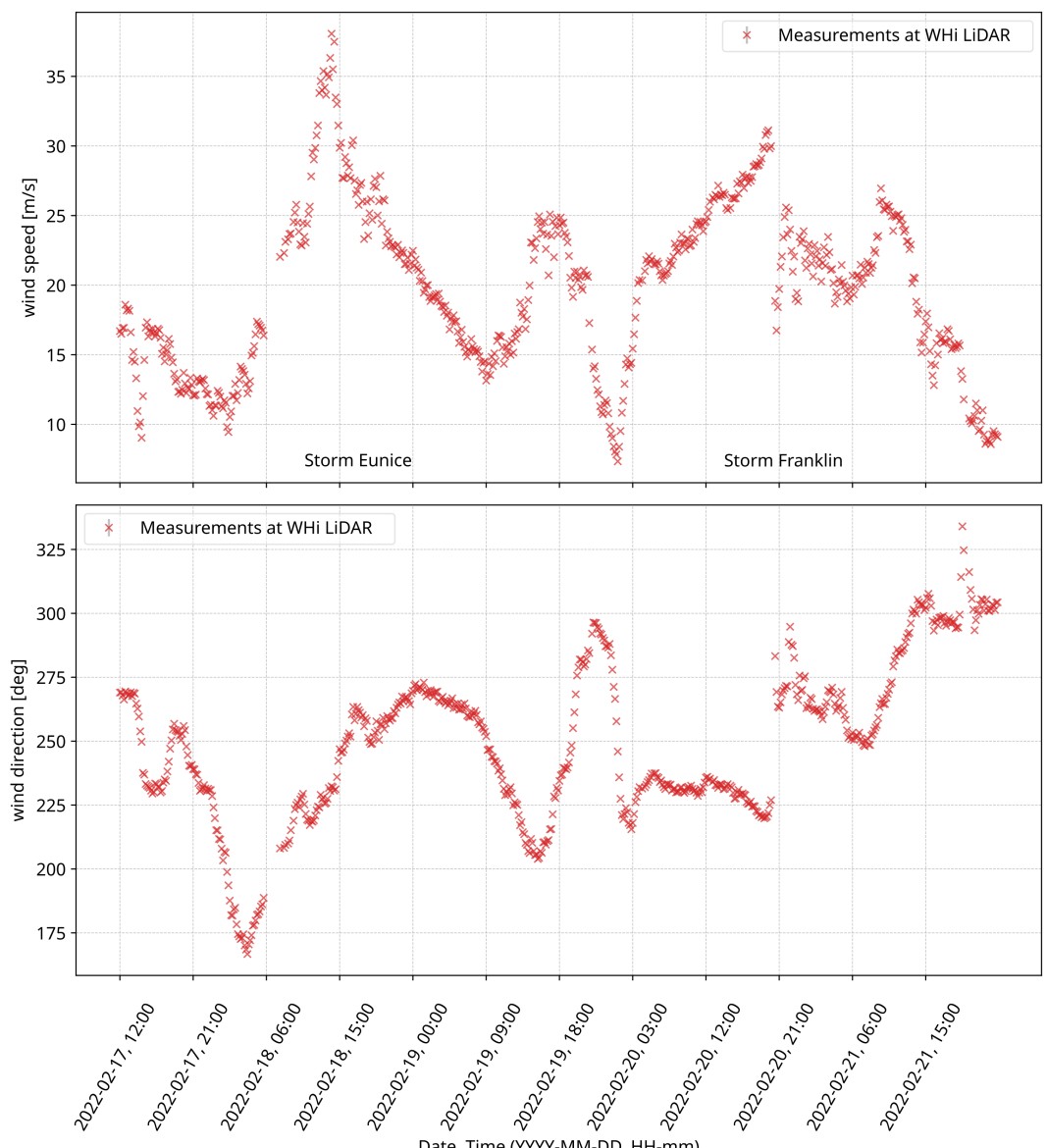

**Figure 3.** Time frame F1: LiDAR observations at 104.5 mTAW height from the Westhinder (WHi) dataset of Glabeke et al. (2023). UTC timezone.

of four distinct numerical experiments (two baseline simulations and two FDDA configurations). In the case of F3 and F4, we carry out three simulations (two baseline and one with FDDA). Finally, to highlight the benefit of offshore measurement

campaigns, nudging is applied specifically at near-hub height (104.5 m), rather than across the entire LiDAR profile. This approach underscores the value of such offshore campaigns: even if data collection is spatially limited by weather conditions or technical issues, the data can still provide meaningful input for improving model accuracy.

## 2.5 Insight into optimal FDDA practices: numerical experiments in F2

In this section, we describe the sensitivity study conducted during time frame F2, from 17 February 2022 to 18 February 2022. The primary objective of this study is to gain insight into the optimal practices for FDDA, particularly focusing on varying the nudging strength and radius of influence. The specific values of the nudging parameters used in this study are detailed in Table 3.

Within F2, we perform a sensitivity study examining the impact of the radius of influence and the nudging strength in FDDA. The assimilation of upwind LiDAR data is performed for 15 numerical experiments (including one configuration where nudging occurs at consecutive intervals) in which we vary the radius of influence $R_{xy}$ and the nudging strength $G_q$. The assimilation of SCADA is performed for 3 cases (in which we vary only $R_{xy}$). For the SCADA nudging cases (labeled S01-S03), we test values of $R_{xy} = 2$, 4 and 10 km. For the WHi LiDAR nudging cases (labeled L01-L14 and LC04), we assimilate upwind observations at a height of 104.5 m. The tested values of the radius of influence are $R_{xy} = 10$, 20, 30, 40, 50, and 60 km. We remind that the WHi LiDAR is situated 47 km further from the wind farm sites, and that the key aspect is to leverage these upwind observations. In addition, for the LiDAR FDDA cases with a radius of influence $R_{xy}$ of 20 km (as in e.g. Cheng et al. (2017)), as well as 30 km, we consider five values of nudging strength $G_q$: $6 \times 10^{-4}$ s$^{-1}$ (a default value, e.g. in Cheng et al. (2017)), values of $3 \times 10^{-3}$ s$^{-1}$, $6 \times 10^{-3}$ s$^{-1}$, $9 \times 10^{-3}$ s$^{-1}$, and the strongest value of $3 \times 10^{-2}$ s$^{-1}$. This yields 10 numerical experiments. Furthermore, for the nudging strength of $6 \times 10^{-3}$ s$^{-1}$, we consider 4 additional cases with radius of influence of 10, 40, 50, and 60 km.

Finally, we propose a practical routine for hour-ahead predictions in which FDDA of upwind LiDAR data is performed in a consecutive manner. In this routine (labeled 'LC04' in Table 3 and in subsequent figures) upwind LiDAR data is assimilated for one hour and its effect propagates as the simulation runs. Once this one-hour window elapses, the DA ends, and the model continues to run without further assimilation. This leads to prediction improvements solely due to the downwind propagation of advanced wind information induced by the FDDA effect. Given the distance of approximately 47 km from the Westhinder LiDAR to the Belgian-Dutch cluster, this implies an advection time of 20–70 minutes and a lasting effect of the DA after its end. This procedure can be repeated as many times as desired, using WRF restart files to ensure the model has spin-up. Every restart should be considered as a separate forecast simulation initiated at the time of the completion of LiDAR data collection (ideally available in real-time), so in this view, no future data is assimilated in the model. Improvements can be achieved in any area of interest, provided the data source is upwind. In this configuration, this is the case when the wind is from the South-West.

## 3 Results and discussion

This section is dedicated to comparing the results from the simulations at the five locations of interest (the upwind Belgian WHi LiDAR; the Front WTs and Waked WTs at the selected Belgian wind farm; and finally, the two Dutch LiDARs, EPL and LEG). In Sect. 3.1, we discuss the results from the sensitivity study to nudging parameters in F2, and we identify optimal FDDA configurations. These configurations are then applied to F1, F3, and F4, in Sect. 3.2

**Table 3.** All simulations performed within the F2 time frame, with varied nudging strength $G_q$ and horizontal radius of influence $R_{xy}$. S01-03 denote three numerical experiments in which SCADA is assimilated, whereas L01-L14 are simulations with only upwind LiDAR assimilation. In all of these simulations, the assimilation time window over which each observation point is used in the nudging algorithm is $\tau = 0.6667$ hours (40 minutes). Finally, LC04 is the LiDAR consecutive assimilation configuration for hour-ahead predictions, in which $\tau = 0.16667$ hours (10 minutes).

|  | $G_q = 6 \times 10^{-4}\ \mathrm{s}^{-1}$ | $G_q = 3 \times 10^{-3}\ \mathrm{s}^{-1}$ | $G_q = 6 \times 10^{-3}\ \mathrm{s}^{-1}$ | $G_q = 9 \times 10^{-3}\ \mathrm{s}^{-1}$ | $G_q = 3 \times 10^{-2}\ \mathrm{s}^{-1}$ |
|---|---|---|---|---|---|
| $R_{xy} = 2$ km |  |  | S01 |  |  |
| $R_{xy} = 4$ km |  |  | S02 |  |  |
| $R_{xy} = 10$ km |  |  | S03 |  |  |
| $R_{xy} = 10$ km |  |  | L01 |  |  |
| $R_{xy} = 20$ km | L02 | L03 | L04, LC04 | L05 | L06 |
| $R_{xy} = 30$ km | L07 | L08 | L09 | L10 | L11 |
| $R_{xy} = 40$ km |  |  | L12 |  |  |
| $R_{xy} = 50$ km |  |  | L13 |  |  |
| $R_{xy} = 60$ km |  |  | L14 |  |  |

## 3.1 Results from the numerical experiments in F2 on FDDA practices

The focus is on the day-long case of 17 February 2022 (F2 in Table 2), with the goal to study the sensitivity effects of varying the radius of influence $R_{xy}$ and the nudging strength $G_q$ of FDDA, while nudging either LiDAR or SCADA data every 10 minutes. As previously mentioned, Table 3 summarizes all 18 cases of numerical experiments within F2: nudging SCADA with different radius of influence, and nudging LiDAR with different parameters. We remind that for the FDDA of a LiDAR measurement point at $104.5$ m, six values for the radius of influence $R_{xy}$ were tested ($10$ km, $20$ km, $30$ km, $40$ km, $50$ km, and $60$ km), whereas for SCADA FDDA – $2$ km, $4$ km, and $10$ km were considered. FDDA of solely upwind observations (in this case, from the Westhinder LiDAR) allows for advanced wind information to propagate to the wind farms in the Belgian-Dutch cluster for (an order of) 20–70 minutes in advance.

The effect of different radius of influence and nudging strength can be seen in Fig. 4. This figure shows the difference with and without FDDA of upwind LiDAR for three different cases:

- L02 (Fig. 4(a)) with $R_{xy} = 20$ km and with the default (and lowest) nudging strength value $G_q = 6 \times 10^{-4}\ \mathrm{s}^{-1}$,

- L04 (Fig. 4(b)) with $R_{xy} = 20$ km and with a ten times larger nudging strength $G_q = 6 \times 10^{-3}\ \mathrm{s}^{-1}$,

- L14 (Fig. 4(c)) with $R_{xy} = 60$ km and $G_q = 6 \times 10^{-3}\ \mathrm{s}^{-1}$.

Figure 4 shows the importance of the two nudging parameters $R_{xy}$ and $G_q$. Varying their values results in wind field modifications. Figure 4 further illustrates both positive and negative variations in wind speed values within the difference fields on the left, likely attributed to numerical diffusion and advection in the proximity of the nudged region.

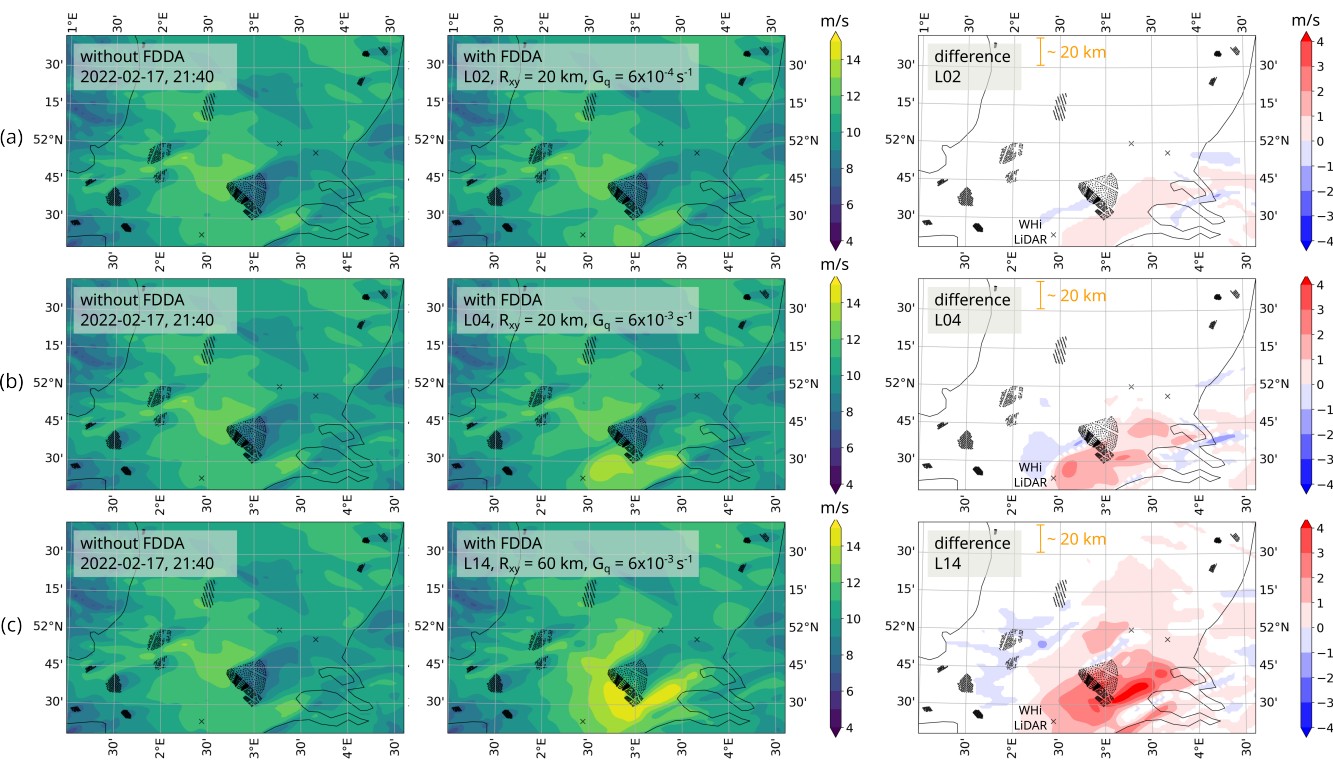

**Figure 4.** Snapshots of wind speed fields (m s$^{-1}$) on 17 February 21:40 UTC, of simulations L02 (a), L04 (b), and L14 (c). In the left columns, the results are without assimilation, whereas in the middle columns FDDA is active. Finally, on the right columns the differences between the two are shown, along with a reference distance length of 20 km.

To explore the performance of the numerical experiments in F2, we compute the MAEs for each measurement height with respect to the WHi LiDAR profile, at the WHi LiDAR location of assimilation (for verification purposes). This is the averaged profile in time frame F2. In Fig. 5, we visualize these MAEs for all different simulations (listed in x-axis) in which LiDAR data is assimilated (see Table 3), as well as the two baseline simulations: a simulation with WFP only (a control run with no FDDA), and a simulation without any WFP. It is indeed expected that MAEs are always reduced at the assimilation location WHi when LiDAR FDDA is performed there. Although the assimilated LiDAR data point is positioned at a measurement height of 104.5 m, we observe enhancements in the entire profile, evident in both wind speed (Fig. 5(a)) and wind direction (Fig. 5(b)). This widespread improvement in height is attributed to the default setting of the vertical radius of influence in FDDA, which spans across all model levels. Hence, the absence of vertical constraints in this influence helps avoid the formation of unusual profiles. Consequently, the assimilation of a LiDAR data point at a single height leads to improvements observed throughout the entire profile. In terms of horizontal influence, improvements are especially pronounced when the horizontal radius of influence $R_{xy}$

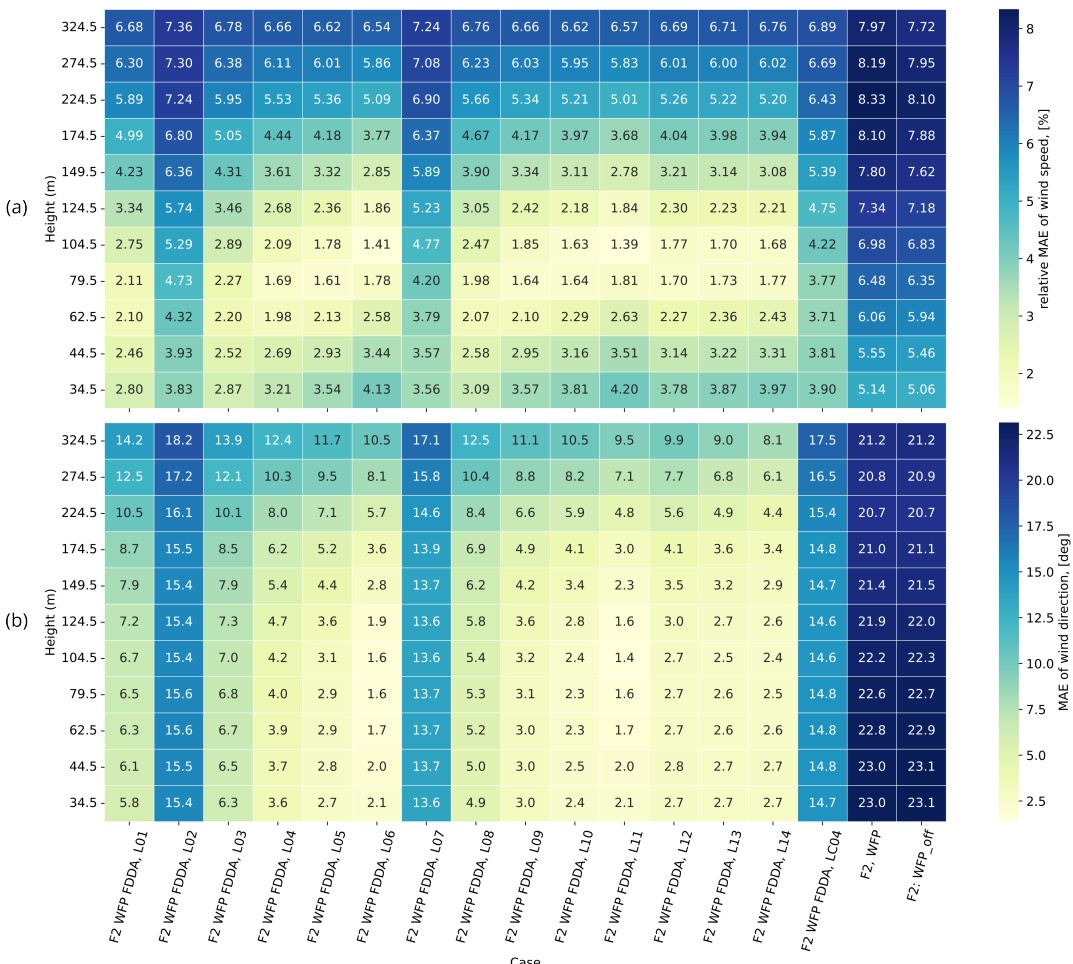

**Figure 5.** MAEs at the WHi assimilation location, computed for each measurement height with respect to the mean profiles from the Westhinder LiDAR observations for simulations in F2.

and the nudging strength $G_q$ are set to higher values (as seen for example in L06, L11). However, when the default value is used for $G_q$ (as in L02, L07) the reduction in MAEs is not as substantial. It is worth remarking also the improvements in the routine in which nudging occurs at consecutive intervals, case 'F2, WFP FDDA, LC04'.

For a full evaluation and determination of optimal FDDA practices, we analyze all experiments within F2, detailed in Table 3, by presenting wind speed RMSEs and biases in Fig. 6 at different locations. Additional error data on wind direction and power is shown in Appendix B. Figure 6 illustrates the numerical results at five locations, benchmarked against local observations. For enhanced clarity, the cells which coincide with an assimilation location have been crossed out, directing the attention to improvements at more distant locations. It is particularly encouraging that the FDDA of the WHi LiDAR point (at a height of $104.5\,\mathrm{m}$) leads to improvements in results $47\,\mathrm{km}$ downwind at turbine sites, outperforming the baseline simulations

without FDDA ('F2: WFP_off' and 'F2, WFP'). When utilizing the default nudging strength value of $6 \times 10^{-4}$ s$^{-1}$, relatively small error reductions are observed downwind at the turbine locations (Front and Waked WTs) in L02 and L07, with L07 outperforming L02 due to its greater horizontal radius of influence (30 km). Similarly, among other pairs with equal nudging strength but differing horizontal radii of influence, L08 surpasses L03, L10 outperforms L05, and L11 excels over L06 due to the greater radius value. The cases L01, L04, L09, L12, L13, L14 share a nudging strength of $6 \times 10^{-3}$ s$^{-1}$, with horizontal radii of influence spanning from 10 km for L01 to 60 km for L14. Interestingly, L09 that has $R_{xy} = 30$ km, demonstrates the most substantial error reduction in this group: increasing the radius beyond 40 km leads to increased biases, as shown in Fig. 6(b). Thus, L04 and L09 (with horizontal radii of 20 and 30 km, respectively) become apparent balanced configurations. Furthermore, while examining varied nudging strengths with a fixed radius $R_{xy}$ of 20 km (L02, L03, L04, L05, L06), and of 30 km (L07, L08, L09, L10, L11), we find consistent RMSE and bias improvements as nudging strength is increased while $R_{xy} = 20$ km. Yet, at 30 km biases worsen despite (inconsistent) RMSE gains. Thus, $R_{xy} = 20$ km is identified as an optimal choice for a horizontal radius of influence. Among L04, L05 and L06, no significant differences are present, which leads to the selection of L04 as the preferred FDDA setting. Therefore, the parameters of L04 and/or L06 are applied in Sect. 3.2 for F1, F3, F4, as well as in a proposed consecutive assimilation routine for F2 in the current section. Finally in Fig. 6, at the more distant EPL and LEG LiDAR comparison locations (approximately 110 km away from the assimilation at WHi LiDAR), wind speed fields remain largely unaffected. At EPL and LEG, a small influence is captured only when the horizontal radius of influence reaches 50 or 60 km.

Figure 7 shows results in the Waked WTs location for three variables: wind speed, wind direction, and power. The wind speed values are again normalized by a cutoff speed (31 m s$^{-1}$). The power is also normalized by a typical rated value (8.4 MW). These results are obtained when nudging only WHi LiDAR upwind. The comparison to SCADA data is at farm sites that are 47 km downwind. When performing FDDA of LiDAR, improvements in results downwind (at the Waked WTs) are evident based on the reduced MAEs in the legends of Fig. 7 for (a) wind speed, (b) wind direction and (c) power. Although the WHi LiDAR is located 47 km away from the wind farm of interest, the wind direction is favorable and from mostly South-West and allows the nudged information to propagate towards the zone of interest (at the Belgian wind farms).

Having upwind observations proves to be especially useful based on the results so far. Overall, the use of nudging shows a significant improvement compared to simulations without it. This methodology can be used as long as local observations are available, but in order to utilize this in a forecasting setting, it is required to understand the behaviour of the FDDA method when the data stops being fed into the simulation. Therefore, to expose the reach of this method, we explore a numerical experiment of FDDA for hour-ahead predictions in which nudging occurs at consecutive intervals that consist of a data assimilation window followed by a forecasting window. Ideally, real-time data access would be a requirement. We nudge the simulation variables closer to observations similarly as in the previous sections, but only during short (one hour) periods of time (nudging windows, abbreviated as 'NW'). Note that these nudging windows are different from the assimilation time window $\tau$, which defines the amount of time for which a single observation is considered in the nudging algorithm ($\tau$ is responsible for the temporal weighing function $w_t$ in $W_q$ of the algorithm). The assimilation is done again at the WHi LiDAR upwind with a nudging radius of influence of 20 km, and with the nudging strength of L04. Figure 8(a) shows the wind speed simulation results for

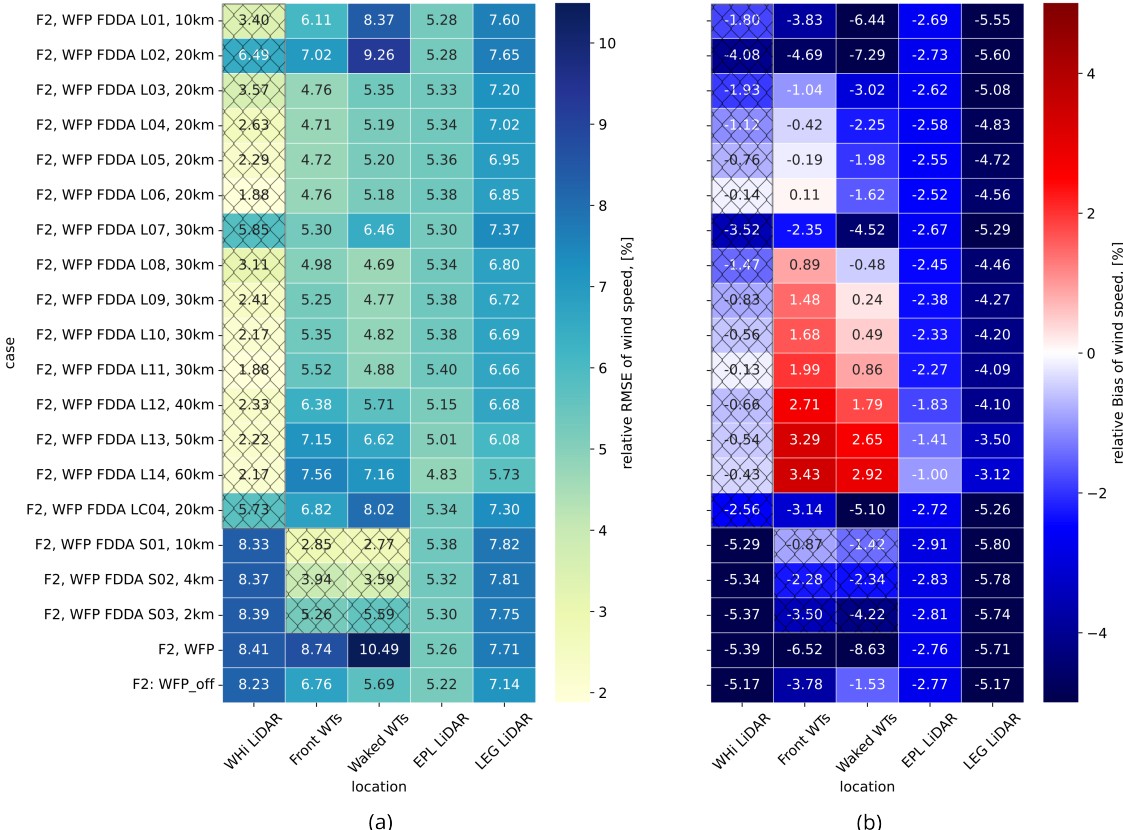

**Figure 6.** Color maps of MAE (a) and bias (b) for wind speed for the different simulations computed at the five locations with respect to the corresponding observations: WHi at 104.5 m, EPL at 116 m, LEG at 115 m, and Front and Waked WTs at hub height. Results at assimilation locations are marked with crossed-out cells.

F2 (17 February 2022) at the WHi LiDAR while assimilating wind data at a height of 104.5 m in four nudging windows, each with a duration of 1 hour. The first nudging window (NW1) is from 12:00 to 13:00 followed by a forecasting window (FCW1) of 2 hours; the second nudging window (NW2) is from 15:00 to 16:00 followed by another forecasting window (FCW2), and so on. Each new nudging window begins with a restart of the simulation. As expected, at the nudging location, wind speed gets closer to the LiDAR observations within the nudging time. However, we also observe in Fig. 8(b) that those quantities still follow better the SCADA observations downwind, even after the end of the nudging window (in the first hour of all forecasting windows). This is explained by the prominently positioned LiDAR observations with respect to the wind farm from the Belgian-Dutch cluster. This position allows for the assimilated quantities (during the nudging window) to be propagated downwind to the wind farm. This advection time is of the order of one hour, and therefore the wind variables at the wind farm are still influenced after the assimilation has stopped. These lead to improved model output downwind at the waked wind turbines, as indicated for example by the reduced MAE values in Fig. 8 (b) for normalized wind speed: from 0.09 to 0.06.

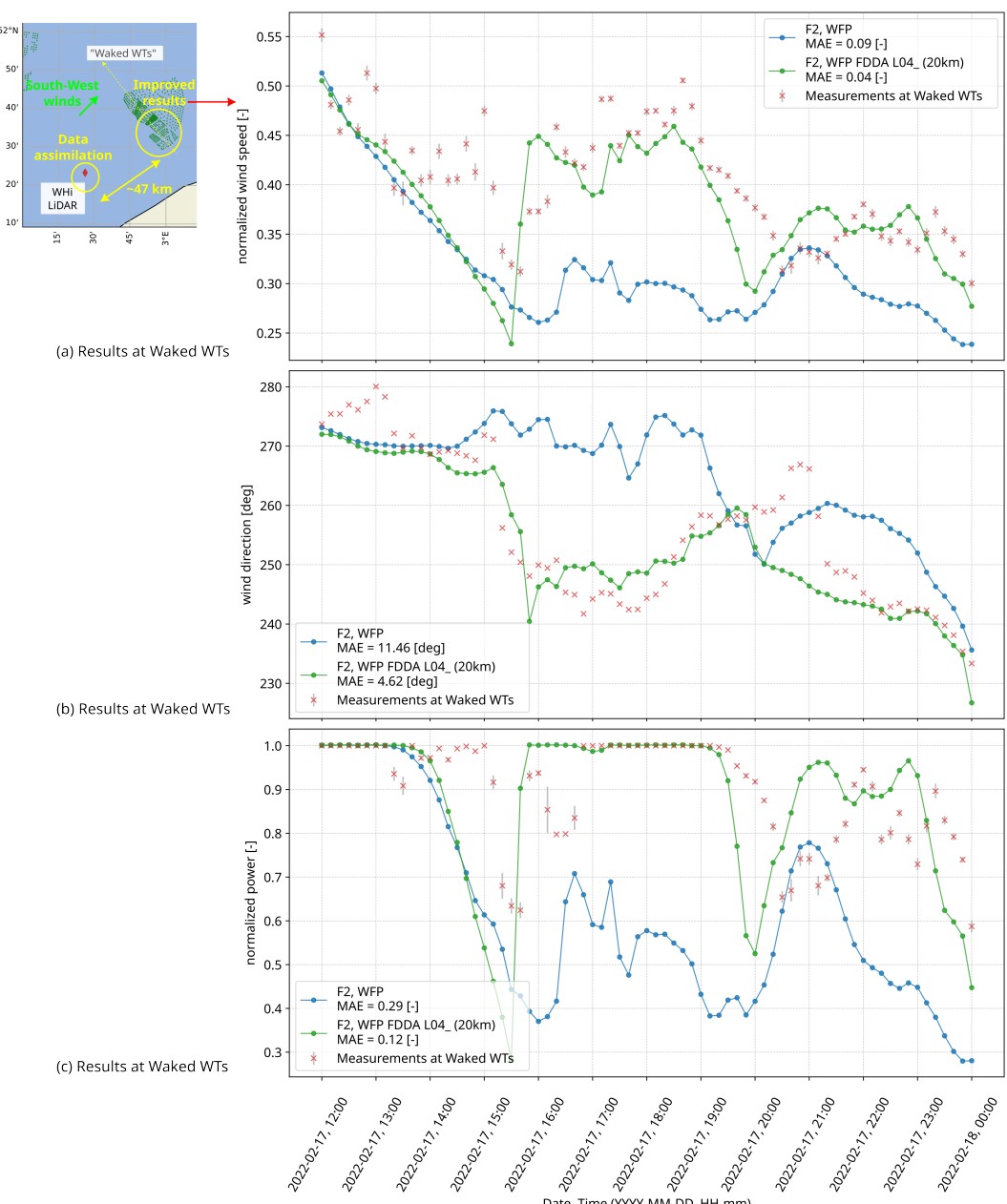

**Figure 7.** Simulation results with and without assimilating upwind WHi LiDAR ('F2, WFP FDDA L04') that are compared to SCADA data from waked WTs. Wind speed (a), wind direction (b), and power (c). Improvements when performing FDDA are highlighted by displaying the MAE values for each variable in the legends. The grey error bars indicate the standard error (available from SCADA only for wind speed and power).

We thus demonstrated a routine with four consecutive nudging and forecasting windows to showcase the potential for hour-ahead improved predictions. A LiDAR that is strategically situated (such as in all of these study cases) can become an essential asset for wind farm decision-making, especially for extreme weather events like a storm, or a frontal passage. Due to the enormous impact that these events might have for wind farm operators, it can be expected that the use of this method will motivate more measurement campaigns offshore, with real-time data access. The main limitation of this strategy is when the flow direction is not from South-West, because we would no longer have LiDAR observations upstream with respect to this direction; in that case, the wind farm is no longer downwind of the nudged quantity, and therefore remains (almost) unaffected.

## 3.2    Results for cases with different weather conditions using WFP and FDDA

We discuss results obtained for time frames from Table 2 (F1, F3, and F4) using the findings from the sensitivity study to nudging parameters in Sect. 3.1. Let us first illustrate results with and without the presence of wind turbines in the computational domain in Fig. 9 for F1 . This figure displays snapshots of wind fields for three arbitrary time slots during this period of interest in February 2022. The wind directions for the three snapshots in Fig. 9 are from South-West (a), South (b), and West (c). Significant velocity deficits are observed in all cases, as well as inter-farm interactions. For the whole duration of the four-day long time frame F1, simulation results at the Waked WTs location are shown in Fig. 10. These results are both for the cases 'F1, WFP' (with active WFP) and 'F1, WFP FDDA L04' (with active WFP and with FDDA of LiDAR located further upwind). The wind speed values are normalized by a representative cut-out speed ($31 \mathrm{~m~s}^{-1}$). We remind that the details on nudging values for L04 are in Table 3. The results using WFP captures well the storms in F1, as well as swift wind direction changes, especially before and during Storm Franklin (19 February 2022 at 18:00, 20 February 2022 at 21:00). In order to further enhance the model output during the three extreme events, we perform FDDA of upwind WHi LiDAR data every 10 minutes for the whole duration of F1, which improves significantly the results of wind speed and wind directions, as indicated by the reduced MAE values for the case 'F1, WFP FDDA' (L04) in Fig. 10.

To evaluate the performance of the different scenarios in wind speed and wind direction results, we present their errors, summarized in Fig. 11. This involves time frames F1, F3, F4, with their corresponding options (without WFP, with WFP, and with FDDA) at the Waked WTs location. The FDDA configuration used is also specified (L04 or L06 from Table 3, the choice of which is supported by the sensitivity study in Sect. 3.1). Moreover, Fig. 11(a) emphasizes that activating WFP helps improve wind speed, but not wind direction. This is the case for F1 and F4, except if the relative bias of wind speed with respect to SCADA data is already negative when WFP is inactive. In that case, the wind power extraction will further bring this bias to more negative values, as it is the case for F3 in Fig. 11(c). Therefore, for frame F3, activating WFP does not improve MAE of wind speed as shown in Fig. 11(a). Activating WFP has almost no impact on wind direction MAEs (Fig. 11(b)) and biases (Fig. 11(d)). However, the introduction of the upwind LiDAR FDDA improves both wind speed and wind direction. FDDA of such observations provides enhanced results that are useful for weather reanalysis, as well as for detailed wind resource assessment.

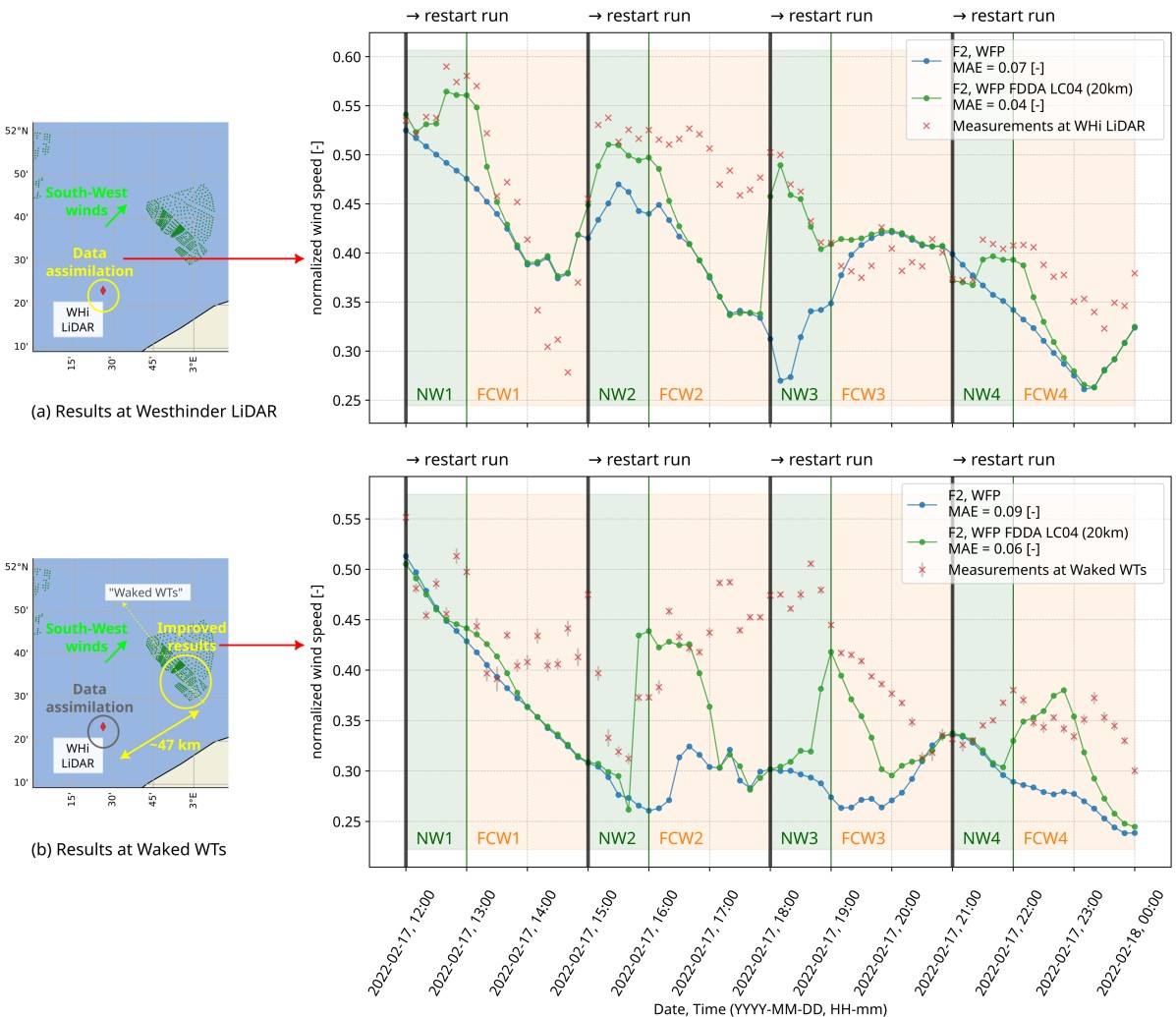

**Figure 8.** Artificial consecutive routine 'F2, WFP FDDA LC04' for hour-ahead predictions: wind speed results at the Westhinder platform (a), compared to the assimilated LiDAR data at a height of 104.5 m in four nudging windows (denoted as NW1-4, one hour each). Information is propagated downwind to the location of the waked WTs, and the results are compared to SCADA (b). Forecasting windows have a 2-hour length and are denoted as FCW1-4. The grey error bars indicate the standard error (available from SCADA only for wind speed and power).

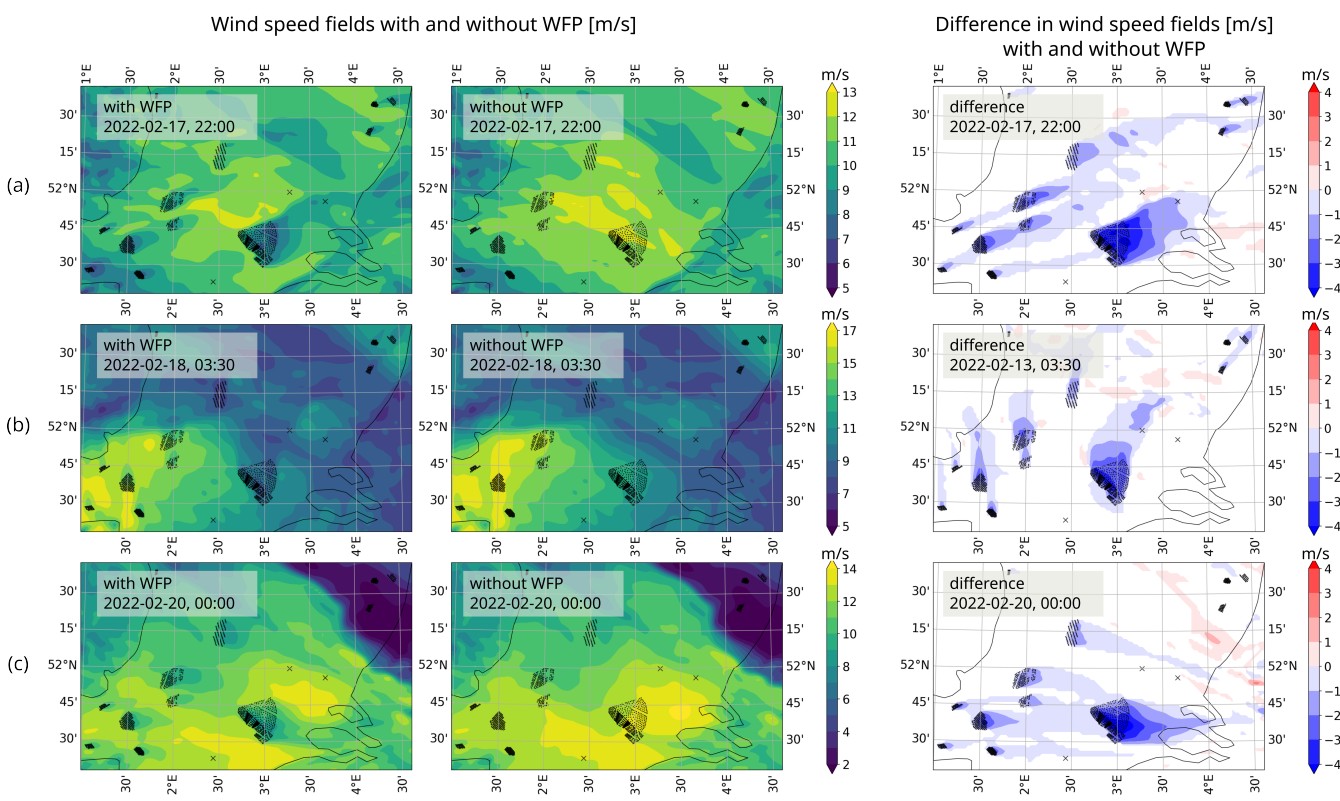

**Figure 9.** Snapshots of wind speed fields $(\mathrm{m\,s^{-1}})$ for three different time instances within F1 (a, b, c). On the left column, results with active WFP show that energy is indeed extracted from the flow. The middle fields are from the simulation without WFP. On the right, the difference in wind speed fields with and without WFP is shown.

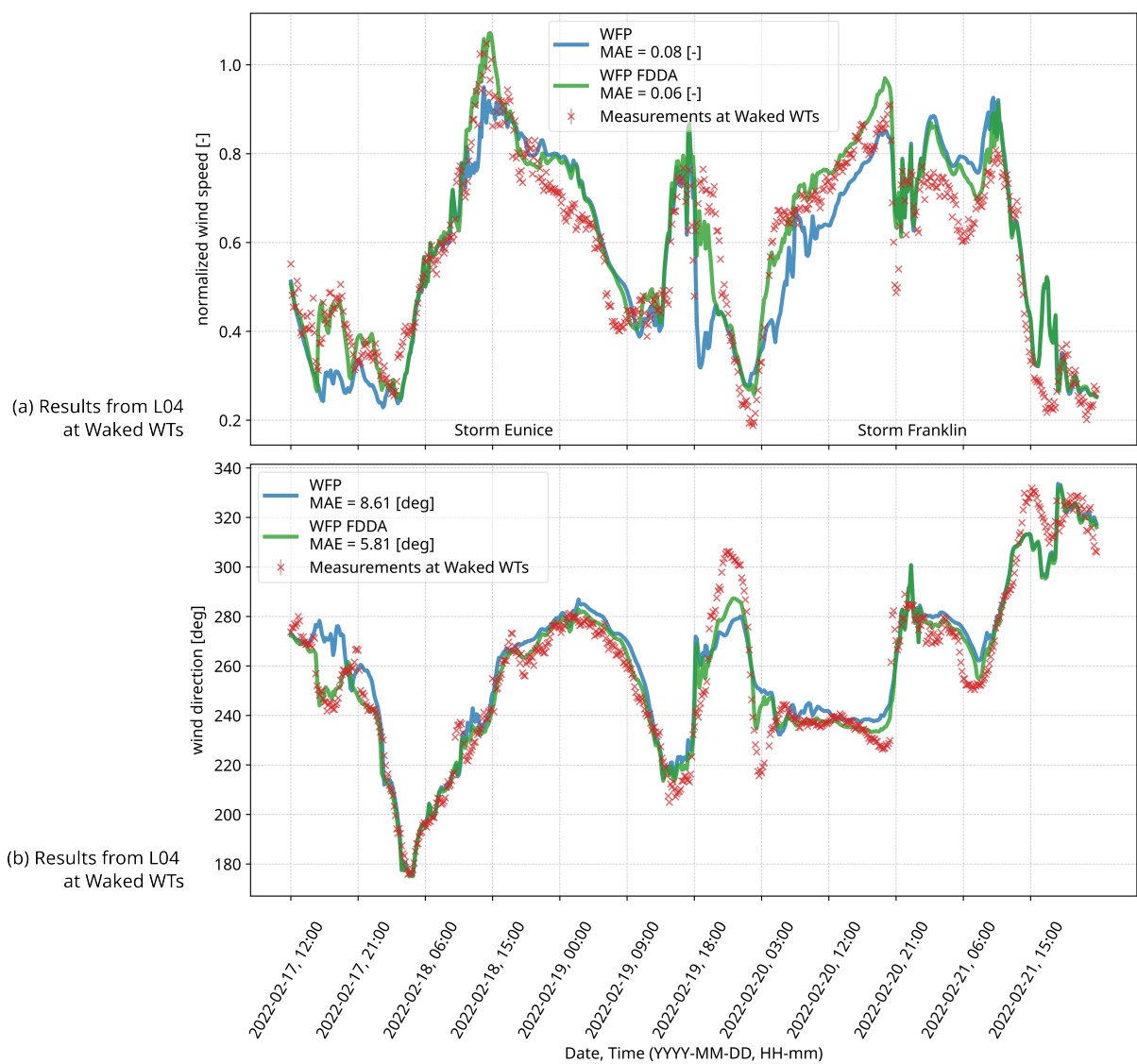

**Figure 10.** The simulations 'F1, WFP' and 'F1, WFP FDDA' (L04) when assimilating (upwind) LiDAR data, 47 km away: results at the Waked WTs location during the Eunice and Franklin storms. UTC timezone.

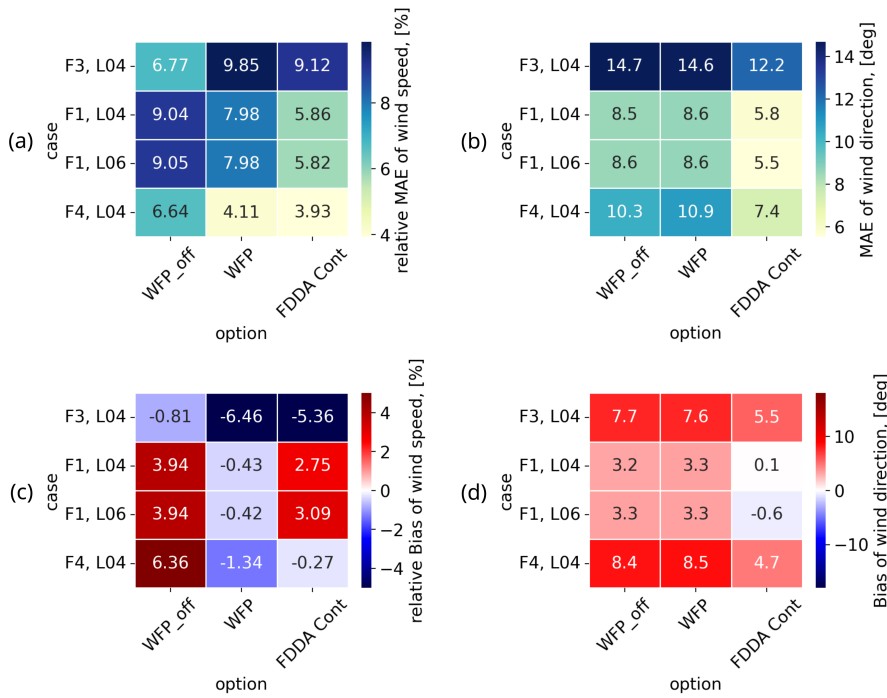

**Figure 11.** Mean absolute errors (MAEs) of wind speed (a) and of wind direction (b), computed for each case at the Waked WTs location. Bias of wind speed (c) and of wind direction (d) for each case at the Waked WTs location. Data is assimilated only at WHi LiDAR (47 km away).

## 4 Conclusions

This study demonstrated the usefulness of assimilating local offshore observations (such as LiDAR) in NWP models (in this case, via the FDDA algorithm in the Advanced Research WRF model) to improve simulation results of wind speed, wind direction and, consequently, power production, based on error reduction during four selected study cases. The simulation results included wind farms in the domain. One of the study cases involved two extreme weather events in February 2022, which were captured well by the simulations. Moreover, we explored the leverage of the FDDA method in a day-long frame by assimilating data either from an upwind (WHi) LiDAR at a specific height, or solely from SCADA at hub height. We performed a sensitivity study via 18 numerical experiments that have eight different values for radius of influence of DA, as well as five different values for nudging strength. This helped to identify an optimal FDDA configuration for this offshore setting. We highlighted the benefits of having an upwind LiDAR, since its assimilation improves results $47\,\mathrm{km}$ downwind at the location of the wind farm. To benefit from this configuration, the only requirement is to have the most common wind direction (which for the Southern Bight of the North Sea is from South-West). The experiments of upwind LiDAR FDDA exhibited improvements in results which were quantified by MAEs, RMSEs, and bias, with respect to the local observations. The identified optimal FDDA setting was also applied to three more study cases. Furthermore, after demonstrating the leverage of upwind nudging, we explored a forecasting routine that contains consecutive nudging windows, which also showed improvements in hour-ahead predictions that were quantified via MAEs.

Limitations of this work include the requirement for a specific range of values for wind direction: the assimilation of West-hinder LiDAR data would not show improvements in the downstream region of interest if the wind direction is not from South-West. Additionally, the lack of offshore observations (especially in real-time) due to the harsh offshore conditions that impact measurement campaigns (as well as cost of deployment, maintenance, structural limitations in deeper waters) reduces the geographical areas in which this method can be applied. Another important limitation is that only prognostic variables can be assimilated with the FDDA method, which is why variational methods are widely used and hold potential for future works. It is worth mentioning that the nudging of waked turbines can affect the physical evolution of turbine and farm wakes at typical mesoscale resolutions that do not resolve individual turbine wakes. A detailed study on this is left for future work. Furthermore, with the increasing density of wind farms installed in the North Sea, the assimilation of SCADA data from neighboring wind farms in NWP models is an important topic for future research.

The methods in this work can be valuable in the future for long-term refined reanalysis (several weeks to few years), where the assimilation of offshore data acquired during wind farm pre-development phase can help reduce bias errors and/or reduce the risk of under-sampling extremes, or where the goal is to evaluate the effects of wind farm decommission on present farms. Practical implications for the wind energy industry can be derived from this research: by utilizing open-source NWP models such as WRF, which is designed for both atmospheric research and operational forecasting applications, more informed wind farm planning and decision-making strategies can be pursued, even under extreme weather conditions. This is especially feasible if offshore measurement campaigns continue to be motivated.

*Code and data availability.* The Advanced Research WRF (ARW) model is developed by the National Center for Atmospheric Research, (Skamarock et al., 2019). WRF v4.5.1 is publicly available at https://github.com/wrf-model/WRF/releases/tag/v4.5.1 (last accessed: September 2023). The forcing data used for initial and boundary conditions in the WRF simulations is also publicly available at the NCEP GFS

0.25 Degree Global Forecast Grids Historical Archive, DOI: 10.5065/D65D8PWK. Data from the numerical simulations and the namelists used in the WRF model are available upon reasonable requests. The postprocessing routines are built using the wrf-python library (Ladwig, 2017). The Westhinder LiDAR data is collected in the framework of the SeaFD project, supported by the Fund for Innovation and Entrepreneurship (VLAIO) at the von Karman Institute for Fluid Dynamics. The LiDAR datasets at the Lichteiland Goeree platform and at the Europlatform are available thanks to the Wind@Sea project, Wind Energy Research Group at TNO Energy Transition (https://www.tno.nl/,

https://nimbus.windopzee.net/). Finally, the SCADA observations of the wind farm of interest, as well as details regarding the wind turbines, are under a non-disclosure agreement (NDA).

## Appendix A:  Included wind farms in the numerical setup

The included wind farms in the numerical setup are described in Table A1, using data from Hoeser et al. (2022) and Hoeser and Kuenzer (2022).

## Appendix B:  Supplementary color maps of errors for wind direction and power in F2

To support the findings on optimal FDDA settings based on wind speed error reduction in Sect. 3.1, we include the errors for wind direction and power. Figure B1 contains wind direction RMSEs and biases that showcase significant improvements when FDDA is performed (especially for the preferred configurations L04 and L06). Figure B2 displays power improvements for RMSE and bias for all FDDA configurations that have an active WFP. These improvements are quite significant, considering

the (almost twice as high) error values in the case 'F2, WFP' (when no data is assimilated). The results in both figures are consistent with the analysis in Sect. 3.1 for wind speed.

**Table A1.** Details on wind farms (lsited in no particular order) in the Southern Bight of the North Sea summarized from Hoeser et al. (2022).

| Wind farm name | Total number of turbines (#) | Turbine Capacity | Hub Height (m) | Rotor Diameter (m) |
|---|---|---|---|---|
| Borssele I Borssele II | 94 | 8.4 MW | 107 | 164 |
| Borssele III Borssele IV | 77 | 9.5 MW | 107 | 164 |
| Borssele V | 2 | 9.5 MW | 107 | 164 |
| Throntonbank I | 6 | 5.0 MW | 93.3 | 126 |
| Throntonbank II & III | 48 | 6.15 MW | 93.3 | 126 |
| Rentel | 42 | 7.35 MW | 102 | 154 |
| Northwind | 72 | 3.0 MW | 80.1 | 112 |
| SeaMade (Seastar) | 30 | 8.4 MW | 107 | 164 |
| Norther | 44 | 8.4 MW | 107 | 164 |
| Nobelwind | 50 | 3.3 MW | 77.1 | 112 |
| Belwind | 55 | 3.0 MW | 70.1 | 112 |
| Belwind Alstom Hailiade | 1 | 6.0 MW | 98.1 | 150 |
| Northwester 2 | 23 | 9.5 MW | 107 | 164 |
| Seamade (Mermaid) | 28 | 8.4 MW | 107 | 164 |
| Scroby Sands | 30 | 2.0 MW | 68 | 80 |
| East Anglia ONE | 102 | 7 MW | 120 | 154 |
| Galloper | 56 | 6 MW | 88 | 154 |
| Greater Gabbard | 140 | 3.6 MW | 78 | 107 |
| Gunfleet Sands | 48 | 3.6 MW | 78 | 107 |
| Gunfleet Sands | 2 | 6 MW | 84 | 120 |
| London Array | 175 | 3.6 MW | 87 | 120 |
| Kentfish Flats | 30 | 3 MW | 70 | 90 |
| Kentfish Flats | 15 | 3.3 MW | 83.6 | 112 |
| Thanet | 100 | 3 MW | 70 | 90 |
| Luchterduinen | 43 | 3 MW | 81 | 112 |
| Egmond aan Zee | 36 | 3 MW | 70 | 90 |
| Princess Amalia | 60 | 2 MW | 60 | 80 |

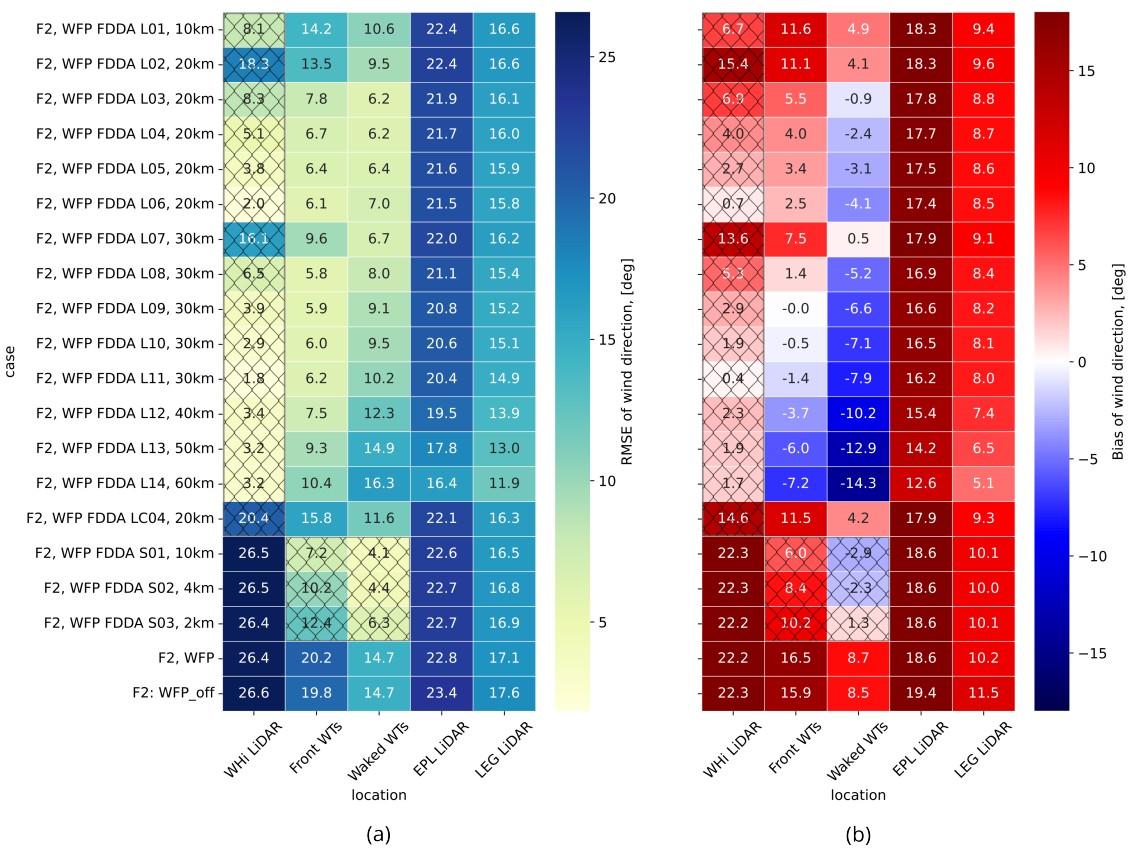

**Figure B1.** Color maps of MAE (a) and bias (b) for wind direction for the different simulations computed at the five locations with respect to the corresponding observations: WHi at 104.5 m, EPL at 116 m, LEG at 115 m, and Front and Waked WTs at hub height. Results at assimilation locations are marked with crossed-out cells.

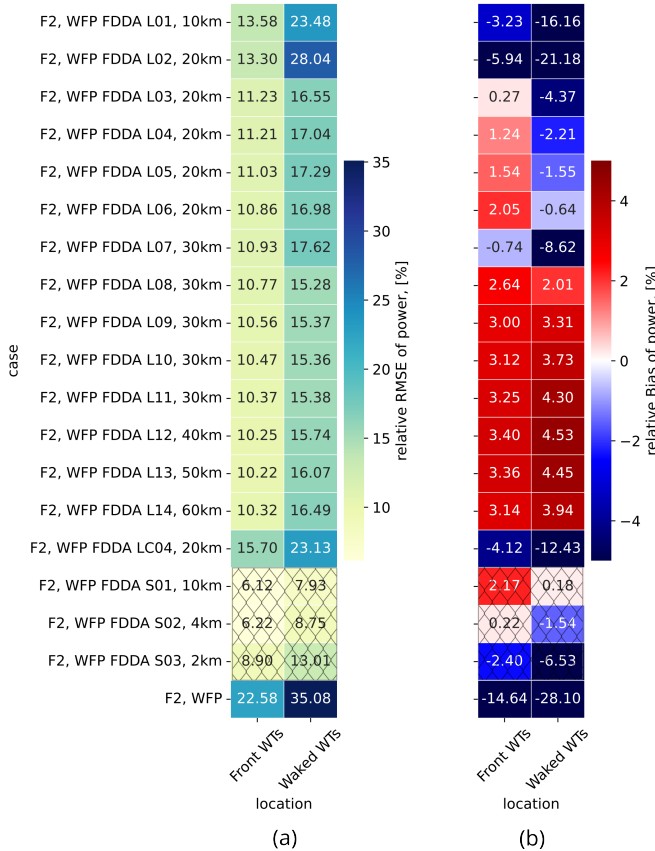

**Figure B2.** Color maps of MAE (a) and bias (b) for power for the different simulations (with an active WFP) computed at the two wind farm locations (Front and Waked WTs) at hub height. Results at assimilation locations are marked with crossed-out cells.

*Author contributions.* Tsvetelina Ivanova: Conceptualization (equal); Data curation (lead); Formal analysis (lead); Investigation (lead); Methodology (lead); Software (lead); Visualization (lead); Writing – original draft preparation (lead); Writing – review & editing (lead).

Sara Porchetta: Formal analysis (equal); Investigation (equal); Methodology (equal); Supervision (equal); Validation (equal); Writing – review & editing (equal).

Sophia Buckingham: Funding acquisition (equal); Methodology (equal); Resources (equal); Writing – review & editing (equal).

Gertjan Glabeke: Investigation (equal); Writing – review & editing (equal).

Jeroen van Beeck: Funding acquisition (equal); Methodology (equal); Resources (equal); Writing – review & editing (equal).

Wim Munters: Conceptualization (lead); Formal analysis (equal); Funding acquisition (lead); Investigation (equal); Methodology (equal); Project administration (lead); Resources (lead); Supervision (lead); Writing – review & editing (equal).

*Competing interests.* The authors declare that no competing interests are present.

*Acknowledgements.* The authors acknowledge the RAINBOW and SeaFD projects, funded by Flanders Innovation & Entrepreneurship (VLAIO) of the Flemish Government. Furthermore, the authors acknowledge the BeFORECAST project, which is supported by the Energy Transition Fund of the Belgian Federal Government. Acknowledgements go towards the Wind@Sea project, Wind Energy Research Group at TNO Energy Transition (https://www.tno.nl/, https://nimbus.windopzee.net/), for the available LiDAR observations at the Lichteiland Goeree platform and at the Europlatform. The authors extend their gratitude to Pieter Mathys from the von Karman Institute for the review and support during the preparation of the manuscript.

The results in this paper are obtained using free and open-source software: the authors extend their gratitude to the communities that have built these powerful tools from which everyone can benefit.

Finally, Tsvetelina Ivanova would like to thank Emmanuel Gillyns for the valuable feedback regarding the manuscript.

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
