# Peer review of "Improving Wind and Power Predictions via Four-Dimensional Data Assimilation in the WRF Model: Case Study of Storms in February 2022 at Belgian Offshore Wind Farms"

_Wind Energy Science, 2023_

## Referee Comment (RC1)

**Review: Improving Wind and Power Predictions via Four-Dimensional Data Assimilation in the WRF Model: Case Study of Storms in February 2022 at Belgian Offshore Wind Farms**

The authors consider a very interesting and important topic: improving wind predictions during extreme events. The authors perform numerical simulations of historical storms in the North Sea that impacted wind turbine clusters. The authors show that employing observational nudging and representing wind turbines using mesoscale wind farm parameterizations improves the model skill in replicating past events. They test sensitivity to nudging parameters in their simulations. The study is interesting and well-written; however, I have major concerns about numerous aspects in the manuscript.

I believe that the most novel aspect of the manuscript is not being fully addressed and most of the paper shows results that are expected based on the methodology. I believe the authors could explore further the importance of having observations that are within multiple advective timescales upstream of the location of interest to improve 6-hr ahead and day-ahead forecasts.

**Major Comments:**

1. The motivation and the results are not connected:

The motivation of this paper, as outlined in the abstract and introduction, is to improve day-ahead and 6-hr ahead forecasting during extreme events. However, the work presented in the paper is not related to forecasting, rather replicating past events by using observational nudging. As the authors clearly explain in the introduction (Lines 44-67), they are nudging the simulations towards the observed state. Therefore, it is not clear how nudging the simulations will improve day/6-hr ahead forecasts, where nudging cannot be performed because the future state of the atmosphere is not known.

The authors briefly investigate how nudging can improve forecasted winds at one location in Sect 3.2 in their "cyclic configuration" (see Major Comment #3 for additional comments on the cyclic approach). However, the authors show their cyclic approach is only useful for 1-hr ahead predictions (depends on the advecting time scale). Furthermore, their cyclic approach also requires observations to repeatedly correct the simulations towards the observed state.

2. Results and methods are not new.

Most of the results are expected based on the methodology and are not new. It is expected that nudging the simulations towards an observed state will improve their skill in representing atmospheric conditions near the observations. It is not clear how observational nudging at every time step will improve forecasts of extreme events.

3. Cyclic configuration:

It is not clear how the cyclic approach that the authors describe in Sect. 3.2 can be used to improve day-ahead or 6-hr ahead forecasts. How is this approach realistic for day-ahead forecasts if you are repeatedly nudging the simulations in a "future" state. The most novel (and most important from my perspective) aspect of Sect. 3.2 is to show that nudging the simulations for a period of time using upstream observations can be useful for forecasting, but only for timescales shorter than the advective time scale.

4. Nudging using SCADA data from waked turbines:

Nudging the WFP simulations using wind speed data from waked turbines modifies both the background flow and wake propagation. However, the authors do not address this in the paper. For instance, as shown in Fig. 8, the simulations without nudging do not capture the observed wind speed and direction during the storm events. And, as shown in Fig. 5, it is likely that the front-row turbines are being waked during the storm event. Thus, nudging the WFP simulations will modify the propagation of wind farm wakes upstream of the turbine locations and artificially change the wake recovery. In a similar manner, nudging the WFP simulations using data from waked turbines inside the wind farm will artificially change wake recovery. Please comment on the limitations of nudging the simulations using data from waked turbines, having in mind that you are not resolving individual turbine wakes with dx=2km.

5. Use of predictions/forecasting throughout the text.

The authors repeatedly use "predictions" or "forecasting" when referring to performing simulations with observational nudging. Please modify this throughout the entire manuscript given that a prediction/forecast refers to estimating a future unknown event, while the authors are nudging simulations using observations.

**Minor Comments:**

1. Line 24 (and rest of text): "Extreme events" is too broad. Please clarify what types of extreme events you are referring to. Some extreme events can occur for winds slower than cut-off (freezing events); others typically don't (e.g., tropical cyclones). Presumably you are most interested in extreme events in which turbines are still operating.
2. Line 35-36: There is still large uncertainty about representing wind farm wakes in mesoscale models using wind farm parameterizations (Eriksson et al., 2017; Peña et al., 2022; Ali et al., 2023). Also, as noted in the studies you highlighted, accurately representing the atmospheric conditions is one of the most important aspects of accurately modeling wind farm wakes. Please mention the uncertainties surrounding mesoscale wind farm parameterizations.

3. Line 116/ Table 1: Please include the value of \alpha (coefficient regulating added TKE in Fitch WFP) used in the simulations.
4. Line 139: Are these lidars vertical profilers? What is their scanning method (DBS, VAD, ...)
5. Line 140: Please show which turbines from the Belgian-Dutch cluster are you using to extract wind speed and power for nudging. (maybe highlight them in Fig. 2)
6. Line 144-145: Are the lidars retrieving data from each beam at 1 Hz? or are the full scans being obtained at 1 Hz (unlikely)?
7. Line 159: Please include basic information on EPL lidar.
8. Sect. 2.3: Please clarify how you define the start and end of each storm.
9. Fig. 7, 11, 12, 13: It is expected that nudging will reduce the MAE at the nudging location. Thus, showing the MAE for the nudging locations is misleading.

**References**

Ali, K., Schultz, D. M., Revell, A., Stallard, T., and Ouro, P.: Assessment of Five Wind-Farm Parameterizations in the Weather Research and Forecasting Model: A Case Study of Wind Farms in the North Sea, Monthly Weather Review, 151, 2333–2359, https://doi.org/10.1175/MWR-D-23-0006.1, 2023.

Eriksson, O., Baltscheffsky, M., Breton, S.-P., Söderberg, S., and Ivanell, S.: The Long distance wake behind Horns Rev I studied using large eddy simulations and a wind turbine parameterization in WRF, J. Phys.: Conf. Ser., 854, 012012, https://doi.org/10.1088/1742-6596/854/1/012012, 2017.

Peña, A., Mirocha, J. D., and Van Der Laan, M. P.: Evaluation of the Fitch Wind-Farm Wake Parameterization with Large-Eddy Simulations of Wakes Using the Weather Research and Forecasting Model, Monthly Weather Review, 150, 3051–3064, https://doi.org/10.1175/MWR-D-22-0118.1, 2022.

---

## Referee Comment (RC2)

wes-2023-177

**Improving Wind and Power Predictions via Four-Dimensional Data Assimilation in the WRF Model: Case Study of Storms in February 2022 at Belgian Offshore Wind Farms**

Tsvetelina Ivanova, Sara Porchetta, Sophia Buckingham, Jeroen van Beeck, and Wim Munters

This manuscript investigates the utilization of offshore in situ observations, specifically from SCADA and lidar devices, to improve wind speed, direction and power prediction of the WRF model near and at a wind farm. The study focuses on the application of an observational nudging approach to assimilate the measurements into WRF and considers also the inclusion of wake effects through the parameterization of wind farms.

**General comments**

- The manuscript is excessively convoluted and lacks clarity, with a presentation of concepts and findings that is somewhat fragmented, making it difficult to follow the logical progression of the study. Authors should consider revising the manuscript readability.
- According to what it stated in the manuscript, the main aim of this study is to improve the forecasting accuracy of a numerical model during extreme events. However, the results and methodology presented do not really address forecasting issues. Instead, they use WRF (past) output information nudged at the same time when (past) observations were done. A reformulation of the objectives and motivation of the paper would better align with the findings presented.
- Although the paper outlines the data assimilation methodology employed, there is a lack of in-depth description regarding the selection criteria for the nudging parameter and the configuration of the nudging algorithm. Key aspects such as vertical and temporal weighting, as well as horizontal nudging weighting function, require proper definition.
- The results section primary outlines the different modelling scenarios considered and presents the accompanying figures and plots. However, there is not enough analysis, interpretation and discussion of the paper's findings. More discussion is required to justify how the paper addresses its objectives. For instance, the discussion provided regarding Fig. 10 provides only a superficial description of the figure itself.
- In situ measurements play a crucial role in the study since they are used for both the assimilation of the WRF model and the validation of the results. However, the description of the measurement devices and corresponding processing is insufficient.
- Finally, it is strongly recommend to align the paper with the specific writing guidelines from WES: WES - Submission (wind-energy-science.net), and to revise the consistency of units, format and styling used. Some examples: using ms$^{-1}$, instead of m s$^{-1}$; using 20h30 instead of 20:30 h, consistency in time format (21h and then 20h00 in page 8), figure captions should be included in the text and not in the figure itself.

**Specific comments:**

Line 20: "…although these can have strengths especially in post-processing techniques". What strengths are you referring to?

Line 86: additional relevant reference: https://doi.org/10.1016/j.jcp.2007.01.037

Lines 93-95: The text enumerates all the parameters of the equation, but it does not provide the values for these parameters used in the study. Moreover, some crucial elements such as the weighting functions (vertical, horizontal and temporal) are not defined. In addition, for clarity, it would be beneficial to provide the description of the nudging methodology subsequent to the description of the WRF model and its configuration.

Line 125: "…with sufficient points across the wind turbine rotor for a typical offshore wind turbine in the Belgian-Dutch cluster." How is the resolution of the WRF model levels within the turbine rotor area? Are these vertical points coinciding with the heights of the measurement devices or there is any vertical interpolation?

Line 133: "… our study investigates different configurations for FDDA with respect to data sources…". In which regard has the FDDA being configuring depending on data sources?

Line 139: Does it refer to a vertical profiler lidar?

Line 142: "The LiDAR (ZX 300M) is installed at that platform since August 2021 and has been collecting wind speed and wind direction information since". It would be beneficial to specify the specific time period during which data was utilized.

Line 143: "…located upstream from the farm…". The use of "upstream" and "downstream" can be ambiguous without a clear reference direction. Consider specifying the direction, such as "southwest" to provide clarity. Idem for the "downstream" used in line 158. Additionally, if only data from the southwest direction were used in the study, it would be relevant to mention this to explain why no wind direction filtering was required.

Line 144: The eleven measurements heights are referenced above the instrument or above the sea level? information on any lidar data processing or quality checks implemented would be valuable.

Line 148: Further details about the SCADA data are necessary. Include information on the quantity of data used, which wind turbines it was collected from, and whether any data processing was conducted. In line 151 it is mentioned that lidar and SCADA data are pre-processed, but these processes are not defined in the manuscript. Then, in line 152 a "standard procedure" to translate to horizontal wind components is mentioned, but not defined.

Line 155: "…LEG platform is positioned at approximately 30 km South-West from Hoek van Holland…". Where is Hoek van Holland?? It is mentioned but not indicated in any map, or specified if it is a region/city/… in (I guess) The Netherlands?

Line 159: It is important to include information about the lidar at the Europlatform. Consider adding details such as the lidar model, measurement heights, and any data processing procedures employed for this particular instrument.

Figure 2: Adding panel labels to figures with multiple panels enhances clarity (applicable to all figures in the paper). In addition, when referring to "two subsets of 5 wind turbines each" in

the figure caption, make sure this concept is introduced and explained in the text prior to the figure. Also, what is the distance between the cluster and the EPL and LEG lidars?

Line 170: "Gradually transient" is unclear and could be interpreted in different ways. Consider revising this phrase to provide a clearer description of the wind flow.

Line 178: "…LiDAR observations in Fig. 3 evaluate…" This is an awkward statement. Instead clarify how the mentioned wind direction (247.52 deg) was calculated.

Line 184: "…contains a subset of 5 wind turbines considered as front row with respect…" Is there any reasoning regarding which 5 turbines are considered? Which specific turbines within the cluster were used?

Figure 3: the top panel plots normalized wind speed, but what is the reference used for normalization? Additionally, it seems there are some data gaps. What is the cause of these gaps?

Line 202: Was it considered to test a larger Rxy? Considering the large distance between the lidar and the turbines, it may be interesting to see if a larger Rxy has a more notorious effect in the predictions at the turbine locations. This would help to elucidate the optimal horizontal weighting configuration of the nudging algorithm, which is one of the paper goals.

Lines 207 to 210: The cyclical approach described needs further clarification regarding its differences and benefits over the standard approach. It is unclear why assimilated measurements for one hour would be more effective than continuous assimilation throughout the period. Additionally, are not the assimilated measurements propagated in the standard approach as well? "…data is assimilated and its effect is propagating as the simulation is running"

Line 219: Including the formula used to calculate MAE would be helpful. Additionally, it is surprising that only one metric (MEA) has been used in the discussion of results. Additionally, while MAE is commonly used, incorporating other metrics like root mean squared error could provide a more comprehensive analysis, especially considering the focus on forecasting where outliers are significant.

Table 2: Clarify the meaning of L1-L6 and S1-S3 in the table caption. Additionally, table captions are usually located above the table, not below.

Line 229: It is mentioned that "The power is also normalized by the rated values.". But no power data is presented in Fig. 6.

Line 233: "…reduced MAE values…". Specify what the reduced MAE values are being compared to for clarity.

Line 239: "…both wind speed and wind direction, especially at the wind farm location.". Is it not this statement somewhat predictable, given that you are comparing the nudged results with the observations used for assimilating WRF?

Line 242: Clarify what is meant by "reanalysis of various events".

Figure 5: Comments regarding the results should go in the main texts, not in the figure caption ("…results… show that energy is indeed extracted from the flow").

Line 245: Why lidar and SCADA data were never assimilated simultaneously?

Line 247: Define what is meant by a "lidar data point". A measurement height maybe?

Line 249: "The configuration of these 11 cases are detailed in Table 2". For clarity and brevity, do not need to repeat this statement since it was already mentioned in line 245.

Figure 6: What are the grey lines next to the scatter crosses indicating? It seems that some scatter points have these lines, and some others don't. Additionally, they are missing in the wind direction plot.

Figure 7: any reasoning about why the assimilation of SCADA is not really improving (or even slightly worsening) the results at the EPL and LEG locations? Considering they are downwards the turbines; shouldn't the propagated nudged information be more productive here? Maybe a larger Rxy would be helpful on this regard?

Line 269: Clarify the difference between the nudging and assimilation windows. The manuscript indicates that "The assimilation window τ during which observations are considered by the model…", but how are the observations being considered here if they are assimilated during nudging window (from line 274: "…quantities being assimilated during the nudging window…").

Line 273: Explain what is meant by the lidar being "prominently positioned".

Fig. 10 is presented but never discussed or commented.

Line 280: "…nudging windows to showcase this effect.". Clarify which effect is being referred here.

Line 287: "…we compute the MAEs in height with respect to the Whi LiDAR profiles.". This sentence is unclear and may require rephrasing or further explanation.

Line 289: "The height of the assimilated LiDAR point is solely at 104.5 m, yet improvements with respect to observations are perceived along the whole profile." . Is this a result of the vertical weighting approach used? It would be clarifying to explicitly describe the weighting approach.

Line 290: "A vertical smoothing in wind speed profiles is expressed in this figure which ensures the smooth transition between simulation and observation." Could you further explain this? How was this smoothing implemented?

Line 305: "Performing FDDA of SCADA also enhances (locally) the predictions (the three rows corresponding to 'F2, WFP FDDA, S1-3' with the three different radius of influence 10 km, 4 km, and 2 km." Could you clarify this sentence?

Figure 8: Recheck date format along the manuscript for consistency.

Figure 9: Provide an explanation for the green shadowed areas in the figure and why they change size along the figure. Caption should include a description of the assimilation window indicated with the tau symbol. Additionally, avoid discussing results in figure captions ("MAE values are reduced for the case of cyclic nudging in 'F2, WFP FDDA L6' as compared to when no FDDA is performed (in 'F2, WFP')").

Figure 11: It seems that increasing the Rxy improves the effect of the FDDA. Why not trying even larger values to find the optimal value of Rxy? Same with Gq.

Figure 12: Discuss why the wind speed MAE of "F2, WFP" is larger than the error of "F2, WFP_off" at all locations.

Line 316: The manuscript primarily discusses the potential of the employed approach for forecasting applications, yet focuses on improving past model data by nudging towards the observation timeframe (also past). Further discussion regarding how the methodology could be adapted for forecasting purposes is needed.

Line 331: "the downstream region of interest if the wind direction is not from South-West". If the wind does not come from South West, the region of interest is not downstream anymore.

Line 332: There are additional and more relevant reasons for the scarcity of offshore in situ data, such as the cost of deployment and maintenance of measurement campaigns, as well as the structural limitations of installing devices in deeper waters.

**Technical comments:**

Line 13: Only cases when the in-text citation is part of the sentence must be formatted as "Skamarock et al. (2019)". In any other case, should be "(Skamarock et al. 2019)". See [WES - Submission (wind-energy-science.net)](WES - Submission (wind-energy-science.net)).

Line 39: "highlights" instead of "highlight"

Line 245: "reference" instead of "referece"

Line 249: "of" instead of "off"

Line 300: "namely when not performing any nudging 'F2, WFP_off' and 'F2, WFP', when nudging only LiDAR data 'F2, WFP FDDA, L1-6's, and when nudging only SCADA 'F2, WFP FDDA, S1-3's, all of which have their parameters detailed in Table 2." This sentence is redundant, since readers are already referred to Table 2.

---

## Author Comment (AC1)

**Response to Reviewers' Comments on WES-2023-177**

Tsvetelina Ivanova, Sara Porchetta, Sophia Buckingham, Gertjan Glabeke, Jeroen van Beeck, Wim Munters

July 7, 2024

We thank both reviewers for their insightful suggestions and remarks, which helped us to improve the manuscript, as well as to revisit, re-frame, and clarify our goals. We believe this has significantly enhanced the quality of our study. In summary, we have complemented our study with additional simulations regarding the sensitivity of the results to nudging parameters (from previously 9, to currently 18 performed simulations) that cover a wider range of values for the nudging strength parameter and the horizontal radius of influence. This supports the goal of identifying the most suitable FDDA practices for this mesoscale setup. We have revised the usage of the terms 'prediction' and 'forecast', and have replaced these with '(simulation) results' or 'model output' where necessary. As suggested by the reviewers, we have adapted our forecasting objective to hour-ahead horizons (approximately the nudged data advection timescale) while clarifying the relevant (cyclic nudging) routine. Furthermore, we have simulated two additional study cases to support our findings, besides the period of Storm Eunice and Storm Franklin (17-22 February 2022). In the remainder of this document, we address each of the reviewer comments point by point. Comments are reported in grey boxes.

**1 Response to Reviewer 1**

> **Overview**
>
> The authors consider a very interesting and important topic: improving wind predictions during extreme events. The authors perform numerical simulations of historical storms in the North Sea that impacted wind turbine clusters. The authors show that employing observational nudging and representing wind turbines using mesoscale wind farm parameterizations improves the model skill in replicating past events. They test sensitivity to nudging parameters in their simulations. The study is interesting and well-written; however, I have major concerns about numerous aspects in the manuscript. I believe that the most novel aspect of the manuscript is not being fully addressed and most of the paper shows results that are expected based on the methodology. I believe the authors could explore further the importance of having observations that are within multiple advective timescales upstream of the location of interest to improve 6-hr ahead and day-ahead forecasts.

**Response:**

We are grateful for the reviewer's positive overview and constructive feedback. We acknowledge the concerns raised and agree that the novelty of our work should be better highlighted and further explored. In response to the reviewer's comment, we have revised our manuscript to more thoroughly address the unique aspects of our study. We therefore focused our efforts on expanding the sensitivity to nudging parameters, as discussed in Section 2.5 of the revised manuscript. We complemented the analysis by testing a wider range of nudging parameters such as larger radius of influence, which allows us to gain better insight into the results at further downstream distances (e.g. LEG and EPL lidars).

Although we agree that including measurements further upstream of the wind farms of interest would be a valuable addition to the study, to the best of our knowledge, no suitable measurements are publicly available to assimilate. In this view, we have reworded the forecasting aspects of the paper to hour-ahead timescales rather than day/6hr-ahead ones.

**Major comment 1**

The motivation and the results are not connected:
The motivation of this paper, as outlined in the abstract and introduction, is to improve day- ahead and 6-hr ahead forecasting during extreme events. However, the work presented in the paper is not related to forecasting, rather replicating past events by using observational nudging. As the authors clearly explain in the introduction (Lines 44-67), they are nudging the simulations towards the observed state. Therefore, it is not clear how nudging the simulations will improve day/6-hr ahead forecasts, where nudging cannot be performed because the future state of the atmosphere is not known.
The authors briefly investigate how nudging can improve forecasted winds at one location in Sect 3.2 in their "cyclic configuration" (see Major Comment 3 for additional comments on the cyclic approach). However, the authors show their cyclic approach is only useful for 1-hr ahead predictions (depends on the advecting time scale). Furthermore, their cyclic approach also requires observations to repeatedly correct the simulations towards the observed state.

**Response:**

We thank the reviewer for this comment. We acknowledge that there was a miscommunication in phrasing our motivation. The primary application of our study is indeed to improve the characterization of wind conditions during past weather events while identifying optimal FDDA practices. The forecasting aspect, as pointed out, is more related to our exploratory work in the cyclic configuration section. In Section 3.2 'Results from the numerical experiments on FDDA practices' (previously named 'FDDA for day-ahead predictions to leverage available upstream observations'), we tested an artificial forecasting routine by employing cyclic nudging of the upstream data (with respect to the most common South-Westerly winds). While this approach currently works for hour-ahead predictions, we agree that it does not directly apply to 6-hour ahead or day-ahead forecasts.

We have revised our manuscript to clarify these points. We have made sure that the motivation is accurately represented and is in line with the results shown in the paper. For example, in the introduction (after Line 71) we have reformulated the main objectives as

- identify the most suitable FDDA practice based on a sensitivity study to nudging parameters

- improve model performance at a wind farm that is approximately 42 km away from the nudging location

- demonstrate the concept of an artificial cyclic configuration with an hour-ahead forecasting potential

**Major comment 2**

Results and methods are not new.
Most of the results are expected based on the methodology and are not new. It is expected that nudging the simulations towards an observed state will improve their skill in representing atmospheric conditions near the observations. It is not clear how observational nudging at every time step will improve forecasts of extreme events.

**Response:**

We understand the concern about the novelty of our results and methods. While it is true that nudging simulations towards an observed state to improve their skill in representing atmospheric conditions is not new, we believe our study presents a unique combination of tools and techniques. Specifically, to our knowledge, no previous study has applied nudging of measurements located kilometers away to improve 1-hour ahead predictions, which can be potentially employed in an operational forecasting setting. This approach, if employed, can aid in the detection and nowcasting of extreme events. However, our study provides additionally a sensitivity study of the results to the nudging parameters, which in turn helps identify best practices for improving wind model output, and allows for hour-ahead predictions. We made sure to highlight these points in our revised manuscript (in Section Introduction) to better articulate the novelty and value of our work.

**Major comment 3**

Cyclic configuration:
It is not clear how the cyclic approach that the authors describe in Sect. 3.2 can be used to improve day-ahead or 6-hr ahead forecasts. How is this approach realistic for day-ahead forecasts if you are repeatedly nudging the simulations in a "future" state. The most novel (and most important from my perspective) aspect of Sect. 3.2 is to show that nudging the simulations for a period of time using upstream observations can be useful for forecasting, but only for timescales shorter than the advective time scale.

**Response:**

We appreciate this feedback on the cyclic approach's limitations for day-ahead or 6-hour ahead forecasts. We agree that it's more suited for shorter timescales (of the order of the advective time scale, approximately one hour ahead). Based on this (and the above) feedback, we have reframed our objectives to focus on short-term forecasts (approximately hour-ahead) and have revised our manuscript accordingly.

For example, we have enhanced Figure 9 with a better description. We perform a simulation as an artificial cyclic routine for hour-ahead forecasting. The simulation is restarted every three hours (while ensuring the utilization of a spinned-up solution). Every restart should be considered as a separate forecast simulation initiated at the time of lidar data collection, so in this view, no future data is ever assimilated in the model. This clarification is added in the revised manuscript after Line 281. During the first hour, LiDAR data is being nudged. The actual forecast then commences after the first hour (once the nudging stops), and is ongoing during a forecasting window of 2 hours, in which we observe the transition from the nudged simulation to the control run. Meanwhile, 40 km away, the effect of the nudging advects, and has an impact on the results during forecasting windows.
* * *
**Major comment 4**

Nudging using SCADA data from waked turbines:
Nudging the WFP simulations using wind speed data from waked turbines modifies both the background flow and wake propagation. However, the authors do not address this in the paper. For instance, as shown in Fig. 8, the simulations without nudging do not capture the observed wind speed and direction during the storm events. And, as shown in Fig. 5, it is likely that the front-row turbines are being waked during the storm event. Thus, nudging the WFP simulations will modify the propagation of wind farm wakes upstream of the turbine locations and artificially change the wake recovery. In a similar manner, nudging the WFP simulations using data from waked turbines inside the wind farm will artificially change wake recovery. Please comment on the limitations of nudging the simulations using data from waked turbines, having in mind that you are not resolving individual turbine wakes with dx=2km.
* * *
**Response:**

We acknowledge that we are not resolving individual turbines with a grid spacing of 2 km, which is a limitation of this study (now noted in the revised manuscript in Section 2 'Methodology and numerical setup'). Indeed by using the actual wind speed measurements in the simulation, we nudge the model variables closer to observations, but potentially alter the wake recovery dynamics.

This clarification is added after Line 417: "It is worth mentioning that the FDDA nudging of waked turbines can affect the physical evolution of turbine and farm wakes at typical mesoscale resolutions that do not resolve individual turbine wakes. A detailed study on this is left for future work."
* * *
**Major comment 5**

Use of predictions/forecasting throughout the text.
The authors repeatedly use "predictions" or "forecasting" when referring to performing simulations with observational nudging. Please modify this throughout the entire manuscript given that a prediction/forecast refers to estimating a future unknown event, while the authors are nudging simulations using observations.
* * *
**Response:**

We agree that the term "prediction" or "forecast" should be used when referring to estimating a future unknown event, whereas "model output" or "simulation results" are more appropriate terms when referring to past events, reanalysis, hindcasts.

We apologize for any confusion caused by our previous (mis-)usage of these terms. We revised our manuscript to ensure that we use "prediction" and "forecast" only when referring to future unknown events, and "model output" or "(simulation) results" when referring to past events or reanalysis (which is the majority of the results in the paper). We appreciate the attention to this detail and believe it will enhance the clarity of our manuscript.
* * *
**Minor comment 1**

Line 24 (and rest of text): "Extreme events" is too broad. Please clarify what types of extreme events you are referring to. Some extreme events can occur for winds slower than cut-ob (freezing events); others typically don't (e.g., tropical cyclones). Presumably you are most interested in extreme events in which turbines are still operating.

**Response:**

We agree that the term "extreme events" is broad and can encompass a variety of scenarios. We are indeed primarily interested in high wind events during which wind turbines are still operating. Line 26 of the revised manuscript: "Furthermore, extreme events that involve high wind speeds in which wind turbines that are still operating can have a profound impact on wind farm operations, since they can often lead to implications for power generation and grid stability."
* * *
**Minor comment 2**

Line 35-36: There is still large uncertainty about representing wind farm wakes in mesoscale models using wind farm parameterizations (Eriksson et al., 2017; Peña et al., 2022; Ali et al., 2023). Also, as noted in the studies you highlighted, accurately representing the atmospheric conditions is one of the most important aspects of accurately modeling wind farm wakes. Please mention the uncertainties surrounding mesoscale wind farm parameterizations.
* * *
**Response:**

We thank the reviewer for this valuable remark, and we therefore include the suggested references in the Introduction. We incorporate these, and we highlight the shortcomings of WFPs in the following way starting after Line 44:

"However, when representing wind farm wakes in mesoscale models using WFP, uncertainties arise (Eriksson et al., 2017; Peña et al., 2022; Ali et al, 2023). Accurately capturing atmospheric conditions is crucial for modeling wind farm wakes. WFP has limitations, including the need for TKE correction and sensitivity to atmospheric stability. Additionally, WFPs restrict options for planetary boundary layer schemes, which in turn affects the fidelity of boundary layer representation."
* * *
**Minor comment 3**

Line 116/ Table 1: Please include the value of $\alpha$ (coebicient regulating added TKE in Fitch WFP) used in the simulations.
* * *
**Response:**

The coefficient used is that of default setting in WRF, and it is $\alpha = 0.25$. We include this information on Page 4 in the following sentence, as well as in Table 1:

"The performed WFP simulations take into account the TKE advection with a correction factor $\alpha$ of 0.25, following Archer et al. (2020). This coefficient $\alpha$ is denoted in Table 1."
* * *
**Minor comment 4**

Line 139: Are these lidars vertical profilers? What is their scanning method (DBS, VAD, . . . )
* * *
**Response:**

The lidars are indeed vertical profilers (ZX-300M), not a scanning lidar. The measurement principle is similar to a VAD scan with 50 points. We add this information in the manuscript in Section 2.3 'Available offshore observations'. Our lidar specialist provided more details on the lidar configuration and is therefore added as co-author to the current paper.
* * *
**Minor comment 5**

Line 140: Please show which turbines from the Belgian-Dutch cluster are you using to extract wind speed and power for nudging. (maybe highlight them in Fig. 2)
* * *
**Response:**

Unfortunately, due to a non-disclosure agreement related to the SCADA data, we cannot specify the exact turbines of the cluster (nor the farm they are in). In this work, we discuss results at turbines at the front row with respect to the (most common) wind direction from South-West, as well as waked wind turbines in the middle of that particular Belgian farm.

Line 144-145: Are the lidars retrieving data from each beam at 1 Hz? or are the full scans being obtained at 1 Hz (unlikely)?

**Response:**

Each point on the scan is measured at 50 Hz. The wind vector is a result of this scan (radial wind speeds) and is thus streamed at 1 Hz. This is added in the revised manuscript starting Line 159:

"The LiDAR is retrieving wind vector data at a frequency of 1 Hz per height. A full measurement of the vertical profile typically takes 17 seconds due to the extra time for beam focus adjustment, as well as for additional weather condition measurements used in quality control. Thus, for a 10-minute interval, the maximum number of validated wind speed and wind direction measurements is 35. The validation is based on a wind-industry filtering, performed by the LiDAR software (User's Manual ZephIR 2018), as meteorological conditions can result in unvalidated wind data (for example, due to low cloud ceiling, fog, or precipitation)."

Line 159: Please include basic information on EPL lidar.

**Response:**

The basic information and EPL lidar is now included in Section 2.3 Available offshore observations, after Line 199

"We perform comparisons of simulations with one more LiDAR ZX-300M wind profiler dataset that is collected at the Europlatform (EPL) . . . . The measurement heights of the EPL LiDAR are 63, 91, 116, 141, 166, 191, 216, 241, 266 and 291 meters. This platform is located in proximity to the LEG platform as indicated in Figure 2(a)."

Sect. 2.3: Please clarify how you define the start and end of each storm.

**Response:**

We consider the storm event around its corresponding peak wind speed. For example, for Storm Eunice, we consider its duration based on the LiDAR measurements, on 18 February 2022 from 12:00 to 18:00. For Storm Franklin, we consider that it is on 20 February 2022 from 15:00 to 21 February 2022 03:00. We add this clarification in the text in Section 2.4 'Case studies of extreme events and different weather conditions' (previously Section 2.3 'Case studies of extreme events at specific locations').

Fig. 7, 11, 12, 13: It is expected that nudging will reduce the MAE at the nudging location. Thus, showing the MAE for the nudging locations is misleading.

**Response:**

It is indeed expected that nudging will reduce the MAE at the nudging location. Nevertheless, we prefer to still show these values for verification purposes, but we have now 'crossed out' the corresponding cells in all figures, in order to avoid this misunderstanding. Furthermore, we clarify doing this in the text, in Section 3.2 'Results from the numerical experiments on FDDA practices'. Detailed changes can be seen in the annotated manuscript.

**2 Response to Reviewer 2**

> **General comment 1**
>
> The manuscript is excessively convoluted and lacks clarity, with a presentation of concepts and findings that is somewhat fragmented, making it difficult to follow the logical progression of the study. Authors should consider revising the manuscript readability.

**Response:**

We appreciate this feedback from the reviewer. We understand that the clarity and logical progression of our manuscript is crucial for readers to follow our study effectively.

We have undertaken a thorough revision of our manuscript. We have refined the introduction and motivation of the paper, while aiming for a better emphasis on the research gaps addressed. We have also worked on enhancing the logical flow of our study to make it easier for readers to follow. We have simplified expressions where possible, and rephrased parts in a more concise way. We thank the reviewer for taking the time to provide us with this constructive feedback.

A detailed description of the changes can be seen in the annotated manuscript. Examples for such changes include:

- In Section 2.3 'Available offshore observations' (previously Section 2.4), we have dedicated a subsection to each of the different observations used in this work (3 LiDARs and the SCADA data)

- Added a descriptive Table 2 of the time frames of interest for the simulations

- Added Table 3 detailing the sensitivity study of nudging parameters

> **General comment 2**
>
> According to what it stated in the manuscript, the main aim of this study is to improve the forecasting accuracy of a numerical model during extreme events. However, the results and methodology presented do not really address forecasting issues. Instead, they use WRF (past) output information nudged at the same time when (past) observations were done. A reformulation of the objectives and motivation of the paper would better align with the findings presented.

**Response:**

We understand the concern on the misalignment between the stated objectives of our study and the methodology and results presented. We acknowledge and agree that we need to rephrase and better emphasize the main points and findings of our work. We appreciate this feedback and believe it is very helpful in improving our manuscript.

While our study does indeed involve the use of past observations, we would like to reframe and clarify our main goals: namely to identify optimal FDDA practices through a sensitivity study to nudging parameters, and to improve short-term forecasts, which are in the order of the advective timescale for the LiDAR data. Please note that these goals are also listed in the reply to Major Comment 1 of Reviewer 1.

It is realistic to have real-time communication of LiDAR data from offshore, therefore when we receive the most recent measurements, we can assimilate them into WRF and run an hour-ahead forecast using the measurements from the previous hour. Doing this periodically is what formulates the cyclic configuration (which should be interpreted as independent forecasts executed at different times, using WRF restart files).

Furthermore, we have taken steps to rename sections of the manuscript to better match the achieved results. More specifically:

- Section 2.4 is now 'Case studies of different weather conditions' (was previously Section 2.3 'Case studies of extreme events at specific locations'). In this section, we describe each of the study cases (time frames F1-4)

- Section 2.5 is now 'Insights into optimal FDDA practices: numerical experiments in F2' (was previously Section 2.4 'Baseline simulations and numerical experiments')

- Section 3.1 is now 'Results for cases with different weather conditions using WFP and FDDA' (was previously Section 3.1 'Forecasting with WFP, and enhancing predictions via FDDA of SCADA for week-ahead simulation')

- Section 3.2 is now 'Results from the numerical experiments in F2 on FDDA practices' (was previously Section 3.2 'FDDA for day-ahead predictions to leverage available upstream observations')

**General comment 3**

Although the paper outlines the data assimilation methodology employed, there is a lack of in-depth description regarding the selection criteria for the nudging parameter and the configuration of the nudging algorithm. Key aspects such as vertical and temporal weighting, as well as horizontal nudging weighting function, require proper definition.

**Response:**

We thank the reviewer for this comment and we add the additional technical description in the new Section 2.2 'The FDDA/nudging algorithm'. Please see the annotated manuscript for detailed changes.

**General comment 4**

The results section primary outlines the different modelling scenarios considered and presents the accompanying figures and plots. However, there is not enough analysis, interpretation and discussion of the paper's findings. More discussion is required to justify how the paper addresses its objectives. For instance, the discussion provided regarding Fig. 10 provides only a superficial description of the figure itself.

**Response:**

We agree with this important remark and we add further discussion of the figures in the manuscript. We added more interpretation and discussion to highlight how the paper addresses its objectives. For example, we have included more snapshots in Figure 7 (previously labeled as Figure 10) which display differences with and without FDDA in wind speed fields. These snapshots show two different values of the nudging strength parameter, as well as two different radii of horizontal influence of the nudging algorithm. The results are discussed in Section 3.2, 'Results from the numerical experiments in F2 on FDDA practices', more specifically starting Line 333 of the revised manuscript:

"Figure 7 shows the importance of the two nudging parameters $R_{xy}$ and $G_q$. Their larger values imply a better match of the model with observations at the assimilation location, which artificially modifies the wind speed field. In turn, these changes need to be compensated to preserve conservation laws, which is why we observe both positive and negative changes of wind speed values in the right of Fig. 7."

**General comment 5**

In situ measurements play a crucial role in the study since they are used for both the assimilation of the WRF model and the validation of the results. However, the description of the measurement devices and corresponding processing is insufficient.

**Response:**

We apologize that our initial description of the measurement devices and their processing was not sufficient. We have revised our manuscript to include a more detailed description of these measurement devices in Section 2.3, 'Available offshore observations', such as scanning methods, frequency of data collection, and reasons for data gaps. This is in line with Minor comments 6, 7, and 8, from Reviewer 1.

**General comment 6**

Finally, it is strongly recommend to align the paper with the specific writing guidelines from WES: WES - Submission (wind-energy-science.net), and to revise the consistency of units, format and styling used. Some examples: using ms-1, instead of m s-1; using 20h30 instead of 20:30 h, consistency in time format (21h and then 20h00 in page 8), figure captions should be included in the text and not in the figure itself.

**Response:**

We acknowledge the inconsistencies pointed out, as well as the need to include figure captions in the text rather than in the figure itself. We thoroughly revise our manuscript to correct these inconsistencies and ensure that it aligns with the WES submission guidelines.

**Minor comment 1**

Line 20: "...although these can have strengths especially in post-processing techniques". What strengths are you referring to?

**Response:**

We were referring to the strengths of statistical postprocessing techniques that can for example correct an ensemble forecast with respect to known biases. After revising the manuscript we find this phrasing redundant and we therefore remove it.

**Minor comment 2**

Line 86: additional relevant reference: https://doi.org/10.1016/j.jcp.2007.01.037

**Response:**

We thank the reviewer for pointing out this relevant reference and we thus include it in Section 2 'Methodology and numerical setup' as:

"The NWP model employed in this work is the Advanced Research WRF (ARW) Model (Skamarock & Klemp, 2008), Skamarock et al., 2019) , v4.5.1, which is a state-of-the-art mesoscale NWP system available in the public domain. ..."

**Minor comment 3**

Lines 93-95: The text enumerates all the parameters of the equation, but it does not provide the values for these parameters used in the study. Moreover, some crucial elements such as the weighting functions (vertical, horizontal and temporal) are not defined. In addition, for clarity, it would be beneficial to provide the description of the nudging methodology subsequent to the description of the WRF model and its configuration.

**Response:**

We agree with this suggestion and we have added the technical description in the new Section 2.2 'The FDDA/nudging algorithm'. Please see the annotated manuscript for detailed changes (similarly as our response to General comment 3).

**Minor comment 4**

Line 125: "...with sufficient points across the wind turbine rotor for a typical offshore wind turbine in the Belgian-Dutch cluster." How is the resolution of the WRF model levels within the turbine rotor area? Are these vertical points coinciding with the heights of the measurement devices or there is any vertical interpolation?

**Response:**

Within the turbine rotor area, for the smallest wind turbine within the Belgian side of the Belgian-Dutch cluster, we have 8 vertical levels that span across the rotor. Whereas for the largest wind turbine, we have 15 levels. When postprocessing simulation results at hub height, vertical interpolation at 107m was indeed necessary.

We mark these details in the revised manuscript in the section 2.1 'The WRF model configuration' (Line 111):

"The lowest level is at 6 m, with sufficient points across the wind turbine rotor for a typical offshore wind turbine in the Belgian-Dutch cluster. More specifically, for the smallest wind turbine within the Belgian side of the Belgian-Dutch cluster, there are 8 vertical levels that span across the rotor, whereas for the largest wind turbine, there are 15 levels."

**Minor comment 5**

Line 133: "... our study investigates different configurations for FDDA with respect to data sources...". In which regard has the FDDA being configuring depending on data sources?

**Response:**

This statement has been now rephrased to "...our study investigates different configurations for FDDA, of either LiDAR or SCADA" due to the previously unsuitable choice of words.

**Minor comment 6**

Line 139: Does it refer to a vertical profiler lidar?

**Response:**

Yes, the Westhinder LiDAR (ZX-300M) is a vertical profiling LiDAR. We add this information in the manuscript in Line 156, Section 2.3 'Available offshore observations' (previously enumerated as Section 2.2).

**Minor comment 7**

Line 142: "The LiDAR (ZX 300M) is installed at that platform since August 2021 and has been collecting wind speed and wind direction information since". It would be beneficial to specify the specific time period during which data was utilized.

**Response:**

The data utilized for generating the wind rose in Figure 2(d) is from 4 August 2021 to 18 July 2022. This is now added in the text after the referred Line. For the simulations, LiDAR data at 104.5m is used for each corresponding time frame F1, F2, F3, F4, with their dates detailed in Table 2 'Time frames of interest in this study, with their corresponding goals.' of the new manuscript.

**Minor comment 8**

Line 143: "...located upstream from the farm...". The use of "upstream" and "downstream" can be ambiguous without a clear reference direction. Consider specifying the direction, such as "southwest" to provide clarity. Idem for the "downstream" used in line 158. Additionally, if only data from the southwest direction were used in the study, it would be relevant to mention this to explain why no wind direction filtering was required.

**Response:**

We added the clear reference direction when using "upstream" and "downstream" throughout the manuscript, and we use the terms "upwind" and "downwind" which are more suitable for this case.

**Minor comment 9**

Line 144: The eleven measurements heights are referenced above the instrument or above the sea level? information on any lidar data processing or quality checks implemented would be valuable.

**Response:**

The eleven measurement heights are in mTAW, which is with respect to the average sea level at low tide in Ostend, Belgium, which is approximately 2.3 meters below mean sea level. Extra information on data-processing and filtering has been added to the text in Section 2.3 'Available offshore observations' under 'LiDAR Profiler at the Westhinder platform (WHi)', Line 161 of the revised manuscript: "The measurement heights are in $\mathrm{mTAW}$ (meters Tweede Algemene Waterpassing), which means that the average sea level at low tide in Ostend (Belgium) is used as the zero level. This value for Ostend is $\pm 2.3$ $\mathrm{m}$ with respect to the mean sea level."

**Minor comment 10**

Line 148: Further details about the SCADA data are necessary. Include information on the quantity of data used, which wind turbines it was collected from, and whether any data processing was conducted. In line 151 it is mentioned that lidar and SCADA data are pre- processed, but these processes are not defined in the manuscript. Then, in line 152 a "standard procedure" to translate to horizontal wind components is mentioned, but not defined.

**Response:**

Unfortunately, as previously mentioned, the SCADA data is under an NDA, therefore we cannot specify which turbines it comes from. Regarding the pre-processing, we simply meant a projection onto the model axes. We rephrase the pointed out sentence in the following way, under 'ummary of Assimilation & Comparison Locations' of Section 2.3:

"Before assimilation in WRF, all wind speed and wind direction observations are projected onto the axes aligned with model $U$ and $V$ velocity variables ..."
* * *
**Minor comment 11**

Line 155: "...LEG platform is positioned at approximately 30 km South-West from Hoek van Holland...". Where is Hoek van Holland?? It is mentioned but not indicated in any map, or specified if it is a region/city/... in (I guess) The Netherlands?

**Response:**

We have added the distances in Figure 2a, and we have rephrased the referred sentence in the following way (Line 193 of the revised manuscript):

"The LEG platform collects meteorological observations and is positioned approximately $110\,\mathrm{km}$ away from the Westhinder platform as shown in Figure 2(a)"
* * *
**Minor comment 12**

Line 159: It is important to include information about the lidar at the Europlatform. Consider adding details such as the lidar model, measurement heights, and any data processing procedures employed for this particular instrument.

**Response:**

We agree and we thus add this information about EPL in the revised manuscript, in Section 2.3 under 'LiDAR at the Europlatform (EPL)' (after Line 199 of the revised manuscript).

"We perform comparisons of simulations with one more LiDAR ZX-300M wind profiler dataset that is collected at the Europlatform (EPL) .... The measurement heights of the EPL LiDAR are 63, 91, 116, 141, 166, 191, 216, 241, 266 and 291 meters. This platform is located in proximity to the LEG platform as indicated in Figure 2(a)."
* * *
**Minor comment 13**

Figure 2: Adding panel labels to figures with multiple panels enhances clarity (applicable to all figures in the paper). In addition, when referring to "two subsets of 5 wind turbines each" in the figure caption, make sure this concept is introduced and explained in the text prior to the figure. Also, what is the distance between the cluster and the EPL and LEG lidars?

**Response:**

We have added panel labels to all figures where applicable. We have clarified what is meant under 'subsets of 5 wind turbines' in Section 2.3 under 'SCADA Nacelle Anemometers at Front & Waked WTs' of the revised manuscript. We have added distances in Figure 2a of the revised manuscript. The distance from the zone of interest within the cluster 'Waked WTs' to the EPL lidar is approximately 55 km, and to the LEG lidar - approximately 63 km. Furthermore, the distance from the Westhinder lidar and the EPL is 97km, whereas to the LEG lidar it is 110 km. Finally, the distance from the Westinder lidar to the 'Front WTs' is approximately 42 km, whereas to the 'Waked WTs' it is approximately 47 km.
* * *
**Minor comment 14**

Line 170: "Gradually transient" is unclear and could be interpreted in different ways. Consider revising this phrase to provide a clearer description of the wind flow.

**Response:**

Rephrased to "Storm Eunice ... Wind direction is ranging from 225 to 275 degrees"
* * *
**Minor comment 15**

Line 178: "...LiDAR observations in Fig. 3 evaluate..." This is an awkward statement. Instead clarify how the mentioned wind direction (247.52 deg) was calculated.

**Response:**

Rephrased to "using the LiDAR observations in Fig. 3, the average wind direction is computed as 247.52 degrees (coming from West-Southwest)"
* * *
**Minor comment 16**

Line 184: "...contains a subset of 5 wind turbines considered as front row with respect..." Is there any reasoning regarding which 5 turbines are considered? Which specific turbines within the cluster were used?

**Response:**

The reason why we use exactly 5 wind turbines is because they all fall in the cell of interest in our numerical domain (the cell that corresponds to the lat,lon coordinates of these turbines). Sometimes there are fewer turbines positioned in various cells. Due to the aforementioned NDA, we cannot disclose which specific turbines are used. We have added this explanation in the revised manuscript in Section 2.3 under 'SCADA Nacelle Anemometers at Front & Waked WTs', Line 188:

"However, due to a non-disclosure agreement with the wind farm operator, we are unable to list the exact coordinates of these turbines."
* * *
**Minor comment 17**

Figure 3: the top panel plots normalized wind speed, but what is the reference used for normalization? Additionally, it seems there are some data gaps. What is the cause of these gaps?

**Response:**

In all figures in which we have normalized the wind speed, we have done so by dividing by a (typical) cutoff speed of 31 m/s. This is now mentioned in the revised manuscript.

The main reason for the LiDAR data gaps in Figure 3 are are meteorological conditions (low cloud ceiling, fog, precipitation,...) which influence the quality of the measurement. This is now described in Section 2.3, Line 168.
* * *
**Minor comment 18**

Line 202: Was it considered to test a larger Rxy? Considering the large distance between the lidar and the turbines, it may be interesting to see if a larger Rxy has a more notorious effect in the predictions at the turbine locations. This would help to elucidate the optimal horizontal weighting configuration of the nudging algorithm, which is one of the paper goals.

**Response:**

We thank the reviewer for this suggestion. In the revised manuscript, we have now included 9 more cases (now 18 in total) in which we test the nudging sensitivity. Besides the previous simulations with 10, 20, and 30 km radius of influence, we now test additionally 40, 50, 60 km. The analysis of these results is included in the revised manuscript, in Section 3.2 'Results from the numerical experiments in F2 on FDDA practices'. For example, the discussion of Figure 7 (previously Figure 10) is expanded after Line 328 of the revised manuscript. Similarly for Figure 10 after Line 372. Figures 11, 12, and 13, report RMSEs and biases, instead of the previous MAE results.
* * *
**Minor comment 19**

Lines 207 to 210: The cyclical approach described needs further clarification regarding its differences and benefits over the standard approach. It is unclear why assimilated measurements for one hour would be more effective than continuous assimilation throughout the period. Additionally, are not the assimilated measurements propagated in the standard approach as well? "...data is assimilated and its effect is propagating as the simulation is running"

**Response:**

While it is true that we would obtain a better estimate of the atmospheric state if we continuously perform nudging, the advantage of the cyclic approach is that it can be applied for hour-ahead forecasting of unknown incoming wind conditions (and not only for estimation of past events). A simulation can be started by nudging the data available from the past hour, and then in the present, we can perform an (hour-ahead) forecast. The improvements in this forecasts are solely due to the advection

of the modified wind speed thanks to the already assimilated data upstream (with respect to south-westerly wind direction) during the previous hour. The cyclic configuration proposed is an artificial forecasting routine that restarts a simulation every 3 hours (1 hour of nudging of past data, then 2 hours of forecasting and no nudging; followed by a new restart to nudge a new batch of data in the past hour, thus no future observations are used). We acknowledge that the explanation of this configuration was unclear in our previous manuscript, and we have improved this in its revised version (after Line 344).
* * *
**Minor comment 20**

Line 219: Including the formula used to calculate MAE would be helpful. Additionally, it is surprising that only one metric (MEA) has been used in the discussion of results. Additionally, while MAE is commonly used, incorporating other metrics like root mean squared error could provide a more comprehensive analysis, especially considering the focus on forecasting where outliers are significant.

**Response:**

We thank the reviewer for this remark. We have now dedicated a subsection for the metrics 'Summary of Locations & Performance metrics', under Section 2.3 'Available offshore observations'. In it, we included the MAE formula (as well as RMSE and bias).

Furthermore, we show results for both MAEs and biases in Figure 4 (for all cases in the redefined frames F1,F2,F3,F4), and for RMSE and biases in Figures 11, 12, 13 (for all numerical experiments on F2), which indeed allows a better interpretation of the results. Please refer to the revised manuscript for these changes and results.
* * *
**Minor comment 21**

Table 2: Clarify the meaning of L1-L6 and S1-S3 in the table caption. Additionally, table captions are usually located above the table, not below.

**Response:**

Due to the newly performed simulations, we have renamed these as described in Table 3 of the revised manuscript. Nevertheless, the labeling convention is still consistent with the previous manuscript, namely: L01-L14 refers to only LiDAR assimilated, LC04 refers to the cyclic LiDAR assimilation, and finally S01-S03 refers to only SCADA assimilated. We added this clarification in the table caption, and we re-positioned the captions above all tables as suggested.
* * *
**Minor comment 22**

Line 229: It is mentioned that "The power is also normalized by the rated values.". But no power data is presented in Fig. 6.

**Response:**

That is correct, we therefore remove this remark from the text.
* * *
**Minor comment 23**

Line 233: "...reduced MAE values...". Specify what the reduced MAE values are being compared to for clarity.

**Response:**

We compare the MAE values of nudged simulations to the case of 'F2, WFP', and we clarify this along the revised manuscript.
* * *
**Minor comment 24**

Line 239: "...both wind speed and wind direction, especially at the wind farm location.". Is it not this statement somewhat predictable, given that you are comparing the nudged results with the observations used for assimilating WRF?

**Response:**

Indeed this result is expected. This is why in the revised manuscript, when comparing results at various locations, we cross-out the cells which show improvements at the nudging location, to avoid misinterpretation. This is done in the revised manuscript for Figures 11, 12, 13.

**Minor comment 25**

Line 242: Clarify what is meant by "reanalysis of various events".

**Response:**

This is now rephrased (Line 315 of the revised manuscript): "FDDA of such observations provides enhanced results which are useful for weather reanalysis, as well as for detailed wind resource assessment, especially if performed over a few months."

**Minor comment 26**

Figure 5: Comments regarding the results should go in the main texts, not in the figure caption ("...results... show that energy is indeed extracted from the flow").

**Response:**

resolved

**Minor comment 27**

Line 245: Why lidar and SCADA data were never assimilated simultaneously?

**Response:**

During our study, we did perform such experiments, but for the purposes of this work, we decided to focus on the effect of upstream data assimilation (with respect to the most common wind direction from South-West). Further exploration of that combination may remain for future works.

**Minor comment 28**

Line 247: Define what is meant by a "lidar data point". A measurement height maybe?

**Response:**

Indeed it is meant as a single measurement height: considering data at 104.5 mTAW.

**Minor comment 29**

Line 249: "The configuration of these 11 cases are detailed in Table 2". For clarity and brevity, do not need to repeat this statement since it was already mentioned in line 245.

**Response:**

resolved

**Minor comment 30**

Figure 6: What are the grey lines next to the scatter crosses indicating? It seems that some scatter points have these lines, and some others don't. Additionally, they are missing in the wind direction plot.

**Response:**

For SCADA measurements, these lines (error bars) indicate the standard error (standard deviation of the wind speed divided by the square root of the number of data points in the corresponding time interval). This way we estimate error bars for the wind speed, and similarly, for power, using the SCADA database. However, no such standard deviation of the wind

direction is provided by the SCADA database, therefore we are unable to estimate and plot wind direction error bars. Finally, for the cases where we plot LiDAR data, no error bars are computed (since only 10-minute averages were processed, without a standard deviation).

We have noted this detail in the relevant figure captions.

> ### Minor comment 31
>
> Figure 7: any reasoning about why the assimilation of SCADA is not really improving (or even slightly worsening) the results at the EPL and LEG locations? Considering they are downwards the turbines; shouldn't the propagated nudged information be more productive here? Maybe a larger Rxy would be helpful on this regard?

**Response:**

The reason for this is two-fold. Firstly, when assimilating SCADA we are using a radius of influence of 2, 4 or 10km, due to the high density of measurement points from the wind turbines. Secondly, for this particular case, the wind direction is not optimal, and thus the (more narrow) influence of the data assimilated is propagated on the right side of the EPL and LEG lidars.

> ### Minor comment 32
>
> Line 269: Clarify the difference between the nudging and assimilation windows. The manuscript indicates that "The assimilation window $\tau$ during which observations are considered by the model...", but how are the observations being considered here if they are assimilated during nudging window (from line 274: "...quantities being assimilated during the nudging window...").

**Response:**

The parameter $\tau$ is responsible for the temporal weighing of each data point that is assimilated in the model. The default value of this parameter in the WRF namelist (section &fdda) is 4/6 hours (0.6667 hours, or 40 minutes), and we have used this throughout all simulations, except for the cyclic configuration, in which we reduced this to 1/6 hours (0.1667 hours, or 10 minutes), in order to better visualize the proposed artificial hour-ahead prediction routine in Figure 9. On the other hand, the nudging window is defined by us, as the window during which we feed many data points to the model (for example in the cyclic configuration, that is 1 hour). This is clarified in the revised manuscript after Line 349:

"We nudge the simulation variables closer to observations in a similar manner as in the previous sections, but only during short (one hour) periods of time (nudging windows, abbreviated as 'NW'). Note that these nudging windows are different than the assimilation time window $\tau$, which defines the amount of time for which a single observation will be considered in the nudging algorithm (it is responsible for the temporal weight in $W_q$ of the algorithm)."

> ### Minor comment 33
>
> Line 273: Explain what is meant by the lidar being "prominently positioned".

**Response:**

Positioned upstream with respect to the most common wind direction (SW). Rephrased.

> ### Minor comment 34
>
> Fig. 10 is presented but never discussed or commented.

**Response:**

Figure 7 (previously Figure 10) has been expanded to illustrate snapshots of different nudging strengths and radius of influence. We now expanded its discussion in the revised manuscript.

> ### Minor comment 35
>
> Line 280: "...nudging windows to showcase this effect.". Clarify which effect is being referred here.

**Response:** Rephrased to (Line 364 of the revised manuscript): "We thus demonstrated a cycling routine with four nudging windows to showcase its potential for hour-ahead improved predictions."
* * *
**Minor comment 36**

Line 287: "...we compute the MAEs in height with respect to the Whi LiDAR profiles.". This sentence is unclear and may require rephrasing or further explanation.
* * *
**Response:**

rephrased to "MAEs for each measurement height"
* * *
**Minor comment 37**

Line 289: "The height of the assimilated LiDAR point is solely at 104.5 mTAW, yet improvements with respect to observations are perceived along the whole profile." . Is this a result of the vertical weighting approach used? It would be clarifying to explicitly describe the weighting approach.
* * *
**Response:**

The vertical radius of influence is set to default (0 in eta) in the WRF namelist. While we have not experimented with this value, using the default value helps avoid unusual profiles. We rephrase the referred sentence as follows:

"Although the assimilated LiDAR data point is positioned at a measurement height of $104.5$ m, we observe enhancements in the entire profile, evident in both the wind speed (Fig. 10(a)) and wind direction (Fig. 10(b)). This widespread improvement in height is attributed to the default setting of the vertical radius of influence in FDDA, which spans across all model levels. Hence, the absence of vertical constraints in this influence helps avoid the formation of unusual profiles. Consequently, the assimilation of even a single LiDAR data point can lead to improvements observed throughout the entire profile."
* * *
**Minor comment 38**

Line 290: "A vertical smoothing in wind speed profiles is expressed in this figure which ensures the smooth transition between simulation and observation." Could you further explain this? How was this smoothing implemented?
* * *
**Response:**

We have removed this sentence as its phrasing is indeed not explaining enough. Please refer to the previous Minor comment 37, in which we improved the text:

"'...This widespread improvement in height is attributed to the default setting of the vertical radius of influence in FDDA, which spans across all model levels. Hence, the absence of vertical constraints in this influence helps avoid the formation of unusual profiles. Consequently, the assimilation of even a single LiDAR data point can lead to improvements observed throughout the entire profile."
* * *
**Minor comment 39**

Line 305: "Performing FDDA of SCADA also enhances (locally) the predictions (the three rows corresponding to 'F2, WFP FDDA, S1-3' with the three different radius of influence 10 km, 4 km, and 2 km." Could you clarify this sentence?
* * *
**Response:**

This sentence meant that results are improved at the turbines locations of assimilation of the SCADA. It is now removed as it is redundant, and replaced by the following clarifications:

"For a full evaluation and assessment of best practices based on all FDDA cases within F2 (described in Table 2), we present RMSEs and biases in Fig. 11 for wind speed, Fig. 12 for wind direction, Fig. 13 for power. All these figures show results at the five locations (computed with respect to the corresponding local datasets). For clarity, the cells which coincide with an assimilation location have been crossed out. This allows to focus on improvements further away from the assimilation location."

Figure 11 (previously Figure 12) is elaborated in the revised manuscript after Line 384.

| | $G_q = 6 \times 10^{-4}$ s$^{-1}$ | $G_q = 3 \times 10^{-3}$ s$^{-1}$ | $G_q = 6 \times 10^{-3}$ s$^{-1}$ | $G_q = 9 \times 10^{-3}$ s$^{-1}$ | $G_q = 3 \times 10^{-2}$ s$^{-1}$ |
|---|---|---|---|---|---|
| $R_{xy} = 2$ km | | | S01 | | |
| $R_{xy} = 4$ km | | | S02 | | |
| $R_{xy} = 10$ km | | | S03 | | |
| $R_{xy} = 10$ km | | | L01 | | |
| $R_{xy} = 20$ km | L02 | L03 | L04, LC04 | L05 | L06 |
| $R_{xy} = 30$ km | L07 | L08 | L09 | L10 | L11 |
| $R_{xy} = 40$ km | | | L12 | | |
| $R_{xy} = 50$ km | | | L13 | | |
| $R_{xy} = 60$ km | | | L14 | | |

**Minor comment 40**

Figure 8: Recheck date format along the manuscript for consistency.

**Response:**

resolved

**Minor comment 41**

Figure 9: Provide an explanation for the green shadowed areas in the figure and why they change size along the figure. Caption should include a description of the assimilation window indicated with the tau symbol. Additionally, avoid discussing results in figure captions ("MAE values are reduced for the case of cyclic nudging in 'F2, WFP FDDA L6' as compared to when no FDDA is performed (in 'F2, WFP')").

**Response:**

Figure 9 in the revised manuscript is now refined and improved. The green shadowed areas should not have changed size and are now consistently the same size. We included an explanation of $\tau$ and moved the result discussion away from the figure caption.

**Minor comment 42**

Figure 11: It seems that increasing the Rxy improves the effect of the FDDA. Why not trying even larger values to find the optimal value of Rxy? Same with Gq.

**Response:**

We thank the reviewer for this suggestion. We have further included a wider range for Rxy and Gq, as detailed in the revised manuscript:

Table 3, "All simulations performed within the F2 time frame, with varied nudging strength $G_q$ and horizontal radius of influence $R_{xy}$. S01-03 denote three numerical experiments in which SCADA is assimilated, whereas L01-L14 are simulations with only upwind LiDAR assimilation. ...":

**Minor comment 43**

Figure 12: Discuss why the wind speed MAE of "F2, WFP" is larger than the error of "F2, WFP_off" at all locations.

**Response:**

This is due to the fact that for the selected time frame, the bias is negative (as now shown in Figure 11(b) in the revised manuscript). This means that due to energy extraction at the wind farm locations, this bias will become even more negative. This leads to a larger MAE. For a case with positive bias, the energy extraction will help reduce it and thus will improve the MAE (please refer to the new Figure 4 of the revised manuscript).

**Minor comment 44**

Line 316: The manuscript primarily discusses the potential of the employed approach for forecasting applications, yet focuses on improving past model data by nudging towards the observation timeframe (also past). Further discussion regarding how the methodology could be adapted for forecasting purposes is needed.

**Response:**

We agree with this remark and we have therefore reframe the forecasting goals to only one-hour ahead. This issue was also raised by Reviewer 1 (reference in this document: Reviewer 1, Major comments 1, 3, and 5). In the revised manuscript, we have clarified the potential of the cyclic approach for hour-ahead prediction purposes.

**Minor comment 45**

Line 331: "the downstream region of interest if the wind direction is not from South-West". If the wind does not come from South West, the region of interest is not downstream anymore.

**Response:**

rephrasing resolved

**Minor comment 46**

Line 332: There are additional and more relevant reasons for the scarcity of offshore in situ data, such as the cost of deployment and maintenance of measurement campaigns, as well as the structural limitations of installing devices in deeper waters.

**Response:**

We thank the reviewer for this remark and we clarify this in Section 4. Conclusion.

**Minor comment 1 (technical)**

Line 13: Only cases when the in-text citation is part of the sentence must be formatted as "Skamarock et al. (2019)". In any other case, should be "(Skamarock et al. 2019)". See WES - Submission (wind-energy-science.net).

**Response:**

resolved

**Minor comment 2 (technical)**

Line 39: "highlights" instead of "highlight"

**Response:**

resolved

**Minor comment 3 (technical)**

Line 245: "reference" instead of "referece"

**Response:**

resolved

**Minor comment 4 (technical)**

Line 249: "of" instead of "off"

**Response:**

resolved

**Minor comment 5 (technical)**

Line 300: "namely when not performing any nudging 'F2, WFP_off' and 'F2, WFP', when nudging only LiDAR data 'F2, WFP FDDA, L1-6's, and when nudging only SCADA 'F2, WFP FDDA, S1-3's, all of which have their parameters detailed in Table 2." This sentence is redundant, since readers are already referred to Table 2.

**Response:**

resolved

---

## Referee Report (RR1)

**Review: Improving Wind and Power Predictions via Four-Dimensional Data Assimilation in the WRF Model: Case Study of Storms in February 2022 at Belgian Offshore Wind Farms**

The authors have successfully addressed most of my comments and the manuscript has improved immensely. The manuscript is much more coherent, and the analysis provides a more thorough investigation on the use of observational nudging with the wind farm parameterization for improving hour-ahead forecasts and simulations of past events relevant for wind turbine operations. I have a series of minor comments to clarify different aspects of the manuscript prior to publication.

**Minor Comments:**

1. Methods: Consider reframing the use of five domains to three simulations where the only difference is in the use of the WFP and FDDA in domain d03.
2. It appears you are nudging the simulations using lidar observations from a single height (104.5 m) and the vertical radius of influence will affect the wind profile across all vertical levels in the model. Please clarify why you did not nudge the simulations using the wind measurements from all the lidar-measured heights.
3. Page 8, Line 189: "contained" instead of "located"?
4. Please specify the distance between the last turbine row and the LEG and EPL lidars for the predominant wind direction.
5. Section 2.4: The objective for F1, F3, and F4 seems to be the same. However, the conditions under which the nudging happens may differ (negative wind speed bias in F3, and positive wind speed bias in F4). Please clarify.
6. "Cyclic" routine: Calling it a cyclic routine implies that the ON/OFF nudging procedure has a broader goal. However, as stated in line 383, the cyclic routine showcases the potential for improved hour-ahead forecasts only, there is minimal benefit afterwards. Also, Figure 8 shows that after nudging is deactivated the simulation will inevitably converge to the solution without nudging. Thus, I am not sure this should be framed as a "cyclic" procedure, but rather as exemplifying the improvement in hour-ahead forecasts.
7. Figure 4: Please show the radius of influence in the figure for reference.
8. Page 14, Line 317: Please clarify what you mean by "compensating". Are you implying that the accelerations/decelerations near the radius of influence are due to mass conservation? I would think mass conservations will drive in vertical motions instead. These accelerations/decelerations are likely due to numerical diffusion and advection near the nudged region. Also, flow along a coastline typically displays horizontal gradients in wind speed along the cross-stream direction. So, the accelerations/decelerations within the radius of influence may be explained by the fact that you are nudging spatially using a point measurement.
9. It is worth pointing out that observational nudging may improve the results near the radius of influence, but the large-scale background flow will remain largely unaffected and will still dominate flow evolution far from the nudging location. You clearly show this in Figure 6 (small changes in RMSE and MAE for the LEG and EPL lidars).

---

## Author Response (AR2)

**Response to Reviewers' Comments on WES-2023-177**

Tsvetelina Ivanova, Sara Porchetta, Sophia Buckingham, Gertjan Glabeke, Jeroen van Beeck,
Wim Munters

October 22, 2024

We thank the reviewer for their feedback and for acknowledging improvements in the manuscript. We appreciate the additional minor comments, and we address them here to enhance clarity before publication.

**Response to Reviewer 1**

> **Overview**
>
> The authors have successfully addressed most of my comments and the manuscript has improved immensely. The manuscript is much more coherent, and the analysis provides a more thorough investigation on the use of observational nudging with the wind farm parameterization for improving hour-ahead forecasts and simulations of past events relevant for wind turbine operations. I have a series of minor comments to clarify different aspects of the manuscript prior to publication.

**Response:**

We would like to express our gratitude to the reviewer for their constructive feedback and valuable insights. We are glad to hear that the revised manuscript is considered significantly more coherent and in-depth. In this document, we respond to the reviewer's minor comments, which helps us to further enhance our manuscript.

> **Minor comment 1**
>
> Methods: Consider reframing the use of five domains to three simulations where the only difference is in the use of the WFP and FDDA in domain d03.

**Response:**

We take this remark in consideration, and in Methodology (Line 98) we now further highlight how the three innermost domains differ:

"The model configurations in the three innermost domains (with 2 km grid spacing) differ in the following way: D03 is for simulations without WFP, D04 is for active WFP, and D05 – for active WFP while performing FDDA."

> **Minor comment 2**
>
> It appears you are nudging the simulations using lidar observations from a single height (104.5 m) and the vertical radius of influence will affect the wind profile across all vertical levels in the model. Please clarify why you did not nudge the simulations using the wind measurements from all the lidar-measured heights.

**Response:**

In the beginning of this study, while we were setting up the simulations, we did try numerical experiments in which we would assimilate the whole profile. However, assimilating just a single point provided convincing improvements (due to the lack of constraints in the vertical weighing function of the nudging algorithm, as explained in Line 334 of the revised manuscript). Therefore, to further emphasize the value of upwind point measurements, we proceeded with assimilating only the near-hub height level. We clarify why in Section 2.5, Line 274:

"Finally, to highlight the benefit of offshore measurement campaigns, nudging is applied specifically at near-hub height (104.5 m) rather than across the entire LiDAR profile. This approach underscores the value of such offshore campaigns: even

if data collection is spatially limited by weather conditions or technical issues, the data can still provide meaningful input for improving model accuracy."

**Minor comment 3**

Page 8, Line 189: "contained" instead of "located"?

**Response:**

Resolved (now this is in Line 193).

**Minor comment 4**

Please specify the distance between the last turbine row and the LEG and EPL lidars for the predominant wind direction.

**Response:**

We specify these distances in the text in Sect. 2.3, Lines 201 and 209. They are as follows:

- Waked WTs to LEG: approx. 63 km

- Waked WTs to EPL: approx. 56 km

**Minor comment 5**

Section 2.4: The objective for F1, F3, and F4 seems to be the same. However, the conditions under which the nudging happens may differ (negative wind speed bias in F3, and positive wind speed bias in F4). Please clarify.

**Response:**

Indeed, the objectives for F1, F3 and F4 are very similar. Therefore, we clarify this in Table 2, and we add that their conditions may differ also in Table 2 (in terms of wind speed biases, for example). In Sect. 2.4, we add a short clarification in Line 237.

**Minor comment 6**

"Cyclic" routine: Calling it a cyclic routine implies that the ON/OFF nudging procedure has a broader goal. However, as stated in line 383, the cyclic routine showcases the potential for improved hour-ahead forecasts only, there is minimal benefit afterwards. Also, Figure 8 shows that after nudging is deactivated the simulation will inevitably converge to the solution without nudging. Thus, I am not sure this should be framed as a "cyclic" procedure, but rather as exemplifying the improvement in hour-ahead forecasts.

**Response:**

We thank the reviewer for this observation. The term 'cyclic' was chosen to emphasize that we repeat the routine, in which FDDA can be applied in successive hourly intervals using WRF restart files. Indeed, nudging provides the most significant improvement during the hour-ahead forecast window. The intention behind the 'cyclic' is to highlight the potential for recurring assimilation after each forecasting window. However, we now substitute 'cyclic' with 'consecutive' nudging routine throughout the text, to avoid the presumption that 'cyclic' implies 'going back to the same starting point'. We hope that this choice of terminology is more intuitive.

**Minor comment 7**

Figure 4: Please show the radius of influence in the figure for reference.

**Response:**

In Figure 4, a reference distance of 20 km is now shown in the maps on the right.

**Minor comment 8**

Page 14, Line 317: Please clarify what you mean by "compensating". Are you implying that the accelerations/decelerations near the radius of influence are due to mass conservation? I would think mass conservations will drive in vertical motions instead. These accelerations/decelerations are likely due to numerical diffusion and advection near the nudged region. Also, flow along a coastline typically displays horizontal gradients in wind speed along the cross-stream direction. So, the accelerations/decelerations within the radius of influence may be explained by the fact that you are nudging spatially using a point measurement.

**Response:**

We thank the reviewer for these suggestions. We substitute the mentioned sentence with the following, in Line 325 of the revised manuscript:

"Figure 4 further illustrates both positive and negative variations in wind speed values within the difference fields on the left, likely attributed to numerical diffusion and advection in the proximity of the nudged region."

**Minor comment 9**

It is worth pointing out that observational nudging may improve the results near the radius of influence, but the large-scale background flow will remain largely unaffected and will still dominate flow evolution far from the nudging location. You clearly show this in Figure 6 (small changes in RMSE and MAE for the LEG and EPL lidars).

**Response:**

This is indeed an important conclusion. In the text, we refine Line 360 as follows:

"Finally in Fig. 6, at the more distant EPL and LEG LiDAR comparison locations (approximately 110 km away from the assimilation at WHi LiDAR), wind speed fields remain largely unaffected. At EPL and LEG, a small influence is captured only when the horizontal radius of influence reaches 50 or 60 km."